# Reinforced Lifelong Editing for Language Models

**Zherui Li** [* 1]  **Houcheng Jiang** [* 2]  **Hao Chen** [1]  **Baolong Bi** [3]  **Zhenhong Zhou** [1]  **Fei Sun** [3]
**Junfeng Fang** [4]  **Xiang Wang** [2]

## Abstract

Large language models (LLMs) acquire information from pre-training corpora, but their stored knowledge can become inaccurate or outdated over time. Model editing addresses this challenge by modifying model parameters without retraining, and prevalent approaches leverage hypernetworks to generate these parameter updates. However, they face significant challenges in lifelong editing due to their incompatibility with LLM parameters that dynamically change during the editing process. To address this, we observed that hypernetwork-based lifelong editing aligns with reinforcement learning modeling and proposed **RLEdit**, an RL-based editing method. By treating editing losses as rewards and optimizing hypernetwork parameters at the full knowledge sequence level, we enable it to precisely capture LLM changes and generate appropriate parameter updates. Our extensive empirical evaluation across several LLMs demonstrates that RLEdit outperforms existing methods in lifelong editing with superior effectiveness and efficiency, achieving a **59.24%** improvement while requiring only **2.11%** of the time compared to most approaches. Our code is available at: https://github.com/zhrli324/RLEdit.

## 1. Introduction

Although large language models (LLMs) have achieved significant success in various downstream tasks (Brown et al., 2020; Zhao et al., 2024), their performance is hindered by the outdated or erroneous knowledge they may store, lead-

ing to an ongoing demand for continuous updates to model parameters (Lazaridou et al., 2021; De Cao et al., 2021). A straightforward solution is lifelong model editing, which enables sequential knowledge updates in LLMs without compromising their core performance (Hartvigsen et al., 2023; Wang et al., 2024; Fang et al., 2024). For each knowledge $\mathcal{K}$, conventional lifelong editing methods individually calculate an optimal update, denoted as $\Delta$, to adjust the LLM parameters $\mathcal{W}$ through matrix operation (Meng et al., 2022; 2023), as illustrated in Figure 1 (a). Since the complexity and approximations involved in repetitive matrix operation make conventional methods computationally expensive and error-prone, recently, hypernetwork-based methods (Mitchell et al., 2022a; Tan et al., 2024) have emerged as a more efficient and elegant solution. By modeling the $\mathcal{K} \rightarrow \Delta$ process with a hypernetwork, as shown in Figure 1 (b), the hypernetwork can efficiently map knowledge $\mathcal{K}$ to the required update $\Delta$ once trained on these $\mathcal{K}$-$\Delta$ pairs. This approach eliminates the need for expensive and repetitive matrix operations, thus enabling a more concise and practical model editing process.

However, current hypernetwork-based methods struggle to handle long-term lifelong editing tasks (*e.g.*, tasks involving more than 100 edits) (Yang et al., 2024a;b). To ensure convergence during training, all $\mathcal{K}$-$\Delta$ pairs used to train the hypernetwork must be collected from the same LLM with identical parameters. Consequently, a single hypernetwork can only model the $\mathcal{K} \rightarrow \Delta$ process for a fixed LLM, limiting its applicability across evolving models in lifelong editing scenarios. As illustrated in Figure 1 (b), while locate-then-edit methods like MEMIT (Meng et al., 2023) and AlphaEdit (Fang et al., 2024) can effectively support up to 5,000 edits, hypernetwork-based approaches tend to fail after only around 100 edits. This limitation severely restricts the broader use and further advancement of hypernetwork-based editing approaches.

This leads to a natural question: *"How to keep the hypernetwork effective for lifelong editing?"* To address this, the hypernetwork must: 1) capture dynamic changes in the LLM, and 2) adaptively provide $\Delta$ based on the current LLM. This inspires us to turn to Reinforcement Learning (RL) (Arulkumaran et al., 2017), which aims to: 1) capture dynamic changes in the environment, and 2) adaptively pro-

---

*Equal contribution [1]Beijing University of Posts and Telecommunications [2]University of Science and Technology of China [3]Institute of Computing Technology, Chinese Academy of Sciences [4]National University of Singapore. Correspondence to: Junfeng Fang <fangjf1997@gmail.com>, Xiang Wang <xiangwang1223@gmail.com>.

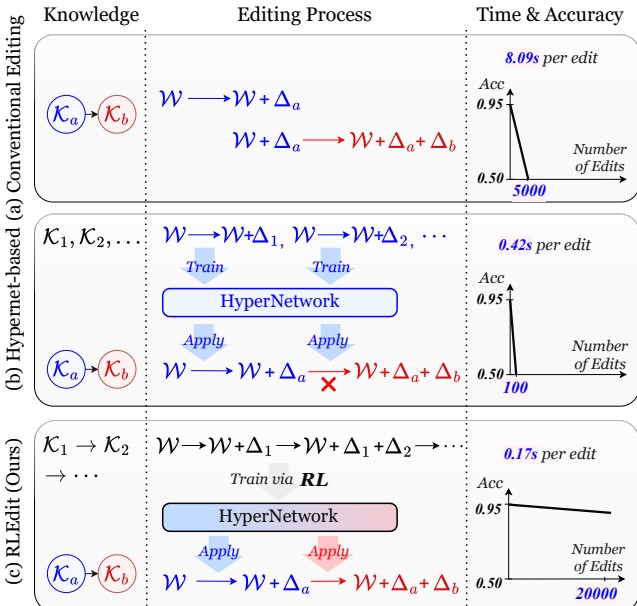

Figure 1: Comparison of different lifelong editing paradigms. Here, $\mathcal{K}_a$ and $\mathcal{K}_b$ represent two consecutive new knowledge samples, while $\Delta_a$ and $\Delta_b$ are their corresponding parameter updates ($\mathcal{K}$ and $\Delta$ with other subscripts are paired data used for training the hypernetwork). (a) requires independent calculations for each knowledge sample, resulting in low efficiency. (b) is efficient but the hypernetwork cannot be generalized to the post-edited LLM. (c) ensures both efficiency and effectiveness, even after more than 20,000 edits. The time shows both initial setup time (including covariance matrix computation or hypernetwork training) and editing time. Best viewed in color.

vide actions based on the current state. Additionally, RL is typically applied in the context of the Markov Decision Process (MDP) (Bellman, 1957). Lifelong editing naturally fits into this framework, as each edit depends solely on the new knowledge and the current LLM state, while the existing loss function and hypernetwork parameters can be modeled as reward and policy respectively.

Building on these observations, we propose **RLEdit**, which applies RL to construct and train the hypernetwork. Specifically, RLEdit treats the hypernetwork as the agent, defines $\Delta$ as the action, and quantifies the performance of LLM as the reward function. We employ an offline policy update approach, enabling RLEdit to recognize the current state of LLMs and adaptively adjust its output $\Delta$, while retaining the efficiency of current hypernetwork-based methods. To meet the requirement that post-edited LLMs must retain previously edited knowledge in lifelong editing, we optimize the reward function by incorporating the difference between the current update and the cumulative sum of previous updates. This modification serves two key purposes: it accelerates

the RL-based training process, and it reduces interference between successive updates.

To further validate RLEdit's capabilities, we conducted extensive experiments on several LLMs. When compared to existing parameter-modifying methods (*e.g.*, RECT (Gu et al., 2024), DAFNet (Zhang et al., 2024b), AlphaEdit (Fang et al., 2024)), RLEdit consistently outperforms these approaches across most evaluation metrics, requiring only 2% of the computation time (including the hypernetwork training overhead) on average. As shown in Figure 1 (c), RLEdit maintains satisfactory performance even after more than 20,000 edits, with each edit taking only 0.17s. Additionally, as the first approach to model lifelong editing as an RL problem, our ablation studies underscore the critical role of the RL framework in RLEdit's performance, paving the way for broader applications of lifelong editing and further advancements in hypernetwork-based methods.

## 2. Preliminary

### 2.1. Lifelong Model Editing

Lifelong model editing requires continuous and sequential model editing on the same LLM, with the total number of edits potentially reaching thousands or even tens of thousands. It requires the post-edited LLM to remember the knowledge from recent edits, retain previously edited knowledge, and maintain comprehensive performance. Let $f_{\mathcal{W}_0} : X \rightarrow Y$ be the initially pre-trained LLM with parameters $\mathcal{W}_0$, which maps the input set X to the output set $Y$. In the lifelong editing task, we have an editing dataset $\mathcal{D}_n = \{(X, Y)|(x_1, y_1), \ldots, (x_n, y_n)\}$, where $X = (x_1, \ldots, x_n)$ is the input stream of knowledge to be edited, and $Y = (y_1, \ldots, y_n)$ is the target output. For the entire input stream $X$, the pre-trained model outputs $f_{\mathcal{W}_0}(X) = Y'$, where $Y' = (y'_1, \ldots, y'_n)$ represents the ground truth. Through sequential editing, we aim to modify the model parameters to achieve $f_{\mathcal{W}_n}(X) = Y$. Therefore, we introduce the model editor (ME). During model editing, the LLM sequentially reads input-output pairs $(x, y)$ from dataset $\mathcal{D}$. When editing the $t$-th knowledge, ME modifies the LLM parameters according to the following formula:

$$f_{\mathcal{W}_t} = \text{ME}(f_{\mathcal{W}_{t-1}}, x_t, y_t), \quad t = 1, \ldots, n. \quad (1)$$

### 2.2. Hypernetwork-based Editing Methods

One approach in model editing is hypernetwork-based editing methods, which involve training an editing hypernetwork to generate LLM parameter updates. Hypernetwork-based methods recognize that LLM fine-tuning gradients contain rich information and use the loss $\mathcal{L}$ of the edited LLM on both edited and unrelated knowledge as the starting point for hypernetwork training. It aims to train a hypernetwork to map fine-tuning gradients to LLM pa-

rameter updates. To address the computational burden of $d \times d$ weight matrices, Mitchell et al. (2022a) decomposes each layer's gradient matrix into a rank-1 product form $\nabla_{\mathcal{W}_l}\mathcal{L} = \delta_{l+1}u_l^\top$, where $\delta_{l+1}$ is the gradient of the loss with respect to pre-activations of layer $l + 1$, and $u_l$ is the input to layer $l$. Through low-rank decomposition, the hypernetwork $\mathcal{H}$ can learn a $d \to d$ mapping:

$$\mathcal{H}: \; \delta_{l+1} \times u_l^\top \to \tilde{\delta}_{l+1} \times \tilde{u}_l^\top, \tag{2}$$

where $\tilde{\delta}_{l+1}$ and $\tilde{u}_l$ are pseudo-activations and pseudo-increments respectively, and the final updates applied to LLM parameters is $\tilde{\nabla}_{\mathcal{W}_l} = \tilde{\delta}_{l+1}\tilde{u}_l^\top$.

In this paper, $\tilde{\nabla}_{\mathcal{W}}$ and $\Delta$ are equivalent, both representing the parameter updates required for editing. We use $\tilde{\nabla}_{\mathcal{W}}$ consistently throughout the rest of this paper.

### 2.3. Reinforcement Learning

A reinforcement learning task is usually formalized as a Markov Decision Process (MDP). An MDP can be defined as a tuple $(\mathcal{S}, \mathcal{A}, \mathcal{R}, \pi, \gamma)$. At each time step $t$, the RL system is in a state $s_t \in \mathcal{S}$, and the agent generates an action $a_t \in \mathcal{A}$ based on its policy $\pi$ and the current state $s_t$, *i.e.*, $a_t \sim \pi(s_t)$. The MDP then transitions to state $s_{t+1}$ and the agent receives a reward $r_t = R(s_t, a_t), r_t \in \mathcal{R}$. After all time steps, the agent optimizes its policy to maximize the total trajectory return $J(\tau) = \sum_{r_t \in \tau} \gamma^t r_t$, where $\gamma$ represents a discount factor and $\tau$ represents one episode trajectory of the agent in the MDP.

## 3. Method

In this section, we elaborate **RLEdit**, a method that achieves efficient and effective lifelong editing through training a hypernetwork adapted to knowledge sequences. We begin by introducing how to establish a reinforcement learning (RL) paradigm for hypernetworks training in lifelong editing (Section 3.1), which includes the key component of RLEdit: reward function design. We then describe RLEdit's training strategy in Section 3.2. Finally, in Section 3.3, we detail the practical implementation and considerations of RLEdit.

### 3.1. Modeling Lifelong Editing as an RL Task

Building an RL paradigm for hypernetworks training in lifelong editing aims to capture dynamic changes in LLM and adaptively generate updates that fit the current parameter state, as mentioned in Section 1. Two key aspects enable this: (1) modeling the hypernetwork training process in lifelong editing as a Markov Decision Process (MDP) (Bellman, 1957; Puterman, 1994), which RL excels at solving (Sutton & Barto, 1998), and (2) carefully designing a reward function tailored to the lifelong editing task.

Table 1: The correspondence between MDP elements and hypernetwork-based lifelong editing components.

| Markov Decision Process | Lifelong Editing |
|---|---|
| Agent | $\mathcal{H}$ |
| Environment | $f_{\mathcal{W}}$ |
| Policy $\pi$ | $\theta$ |
| Action $\mathcal{A}$ | $\tilde{\nabla}_{\mathcal{W}}$ |
| State $\mathcal{S}$ | $(\mathcal{W}, (x, y))$ or $\nabla_{\mathcal{W}}$ |
| Reward $\mathcal{R}$ | $-\mathcal{L}$ |

#### 3.1.1. MDP FORMULATION

The training process of hypernetwork $\mathcal{H}$ in lifelong editing can be formalized as the following procedure:

1. At the $t$-th edit (*i.e.*, time step $t$), new knowledge sample $(x_t, y_t)$ is input to the LLM with parameters $\mathcal{W}_{t-1}$ to collect fine-tuning gradient $\nabla_{\mathcal{W}_{t-1}}$;

2. $\nabla_{\mathcal{W}_{t-1}}$ is fed into $\mathcal{H}$ to get parameter update $\tilde{\nabla}_{\mathcal{W}_t}$;

3. $\tilde{\nabla}_{\mathcal{W}_t}$ is used to update $\mathcal{W}_{t-1}$ to produce $\mathcal{W}_t$;

4. The loss of $(x_t, y_t)$ on the LLM with parameters $\mathcal{W}_t$ is computed to update the hypernetwork.

This process iterates until all knowledge in the training set is traversed. We can observe that the above process only depends on $\mathcal{W}_{t-1}$ and $(x_t, y_t)$ at time step $t$, *i.e.*, it is independent of states from time step 0 to $t - 2$. Therefore, it naturally satisfies the Markov property. Formally:

$$Pr\Big[\mathcal{H}(\nabla_{\mathcal{W}_{t-1}}){=}\tilde{\nabla}_{\mathcal{W}_t}\big|\, \mathcal{H}(\nabla_{\mathcal{W}_{t-2}}){=}\tilde{\nabla}_{\mathcal{W}_{t-1}},$$
$$\ldots, \mathcal{H}(\nabla_{\mathcal{W}_0}){=}\tilde{\nabla}_{\mathcal{W}_1}\Big] \tag{3}$$
$$=Pr\Big[\mathcal{H}(\nabla_{\mathcal{W}_{t-1}}){=}\tilde{\nabla}_{\mathcal{W}_t}\big|\, \mathcal{H}(\nabla_{\mathcal{W}_{t-2}}){=}\tilde{\nabla}_{\mathcal{W}_{t-1}}\Big].$$

To further extend the above Markov process into an MDP, we introduce several concepts: state $s \in \mathcal{S}$, action $a \in \mathcal{A}$, reward $r \in \mathcal{R}$, and policy $\pi$, which are formalized in Table 1. Here, $\theta$ donates the parameters of $\mathcal{H}$ and $\mathcal{L}$ represents the loss on LLM for $(x, y)$. Given $n$ new knowledge samples $(x_i, y_i)_{i=1}^n$ in the training set, the interactions between $s$, $a$, and $r$ over $n$ time steps constitute a trajectory of the MDP:

$$\Big\{(\mathcal{W}_0, (x_1, y_1)), \tilde{\nabla}_{\mathcal{W}_1}, -\mathcal{L}_1, \ldots,$$
$$(\mathcal{W}_{n-1}, (x_n, y_n)), \tilde{\nabla}_{\mathcal{W}_n}, -\mathcal{L}_n\Big\} \tag{4}$$
$$\mapsto \{s_1, a_1, r_1, \ldots, s_n, a_n, r_n\}.$$

More specifically, in each time step, agent $\mathcal{H}$ employs parameters $\theta$ as policy $\pi$ to generate action $\tilde{\nabla}_{\mathcal{W}}$ based on system state $\{\mathcal{W}, (x, y)\}$. This action is then applied to environment $f_{\mathcal{W}}$, resulting in a state transition:

$$\pi(s_t) = a_t, \; s_t + a_t \to s_{t+1}. \tag{5}$$

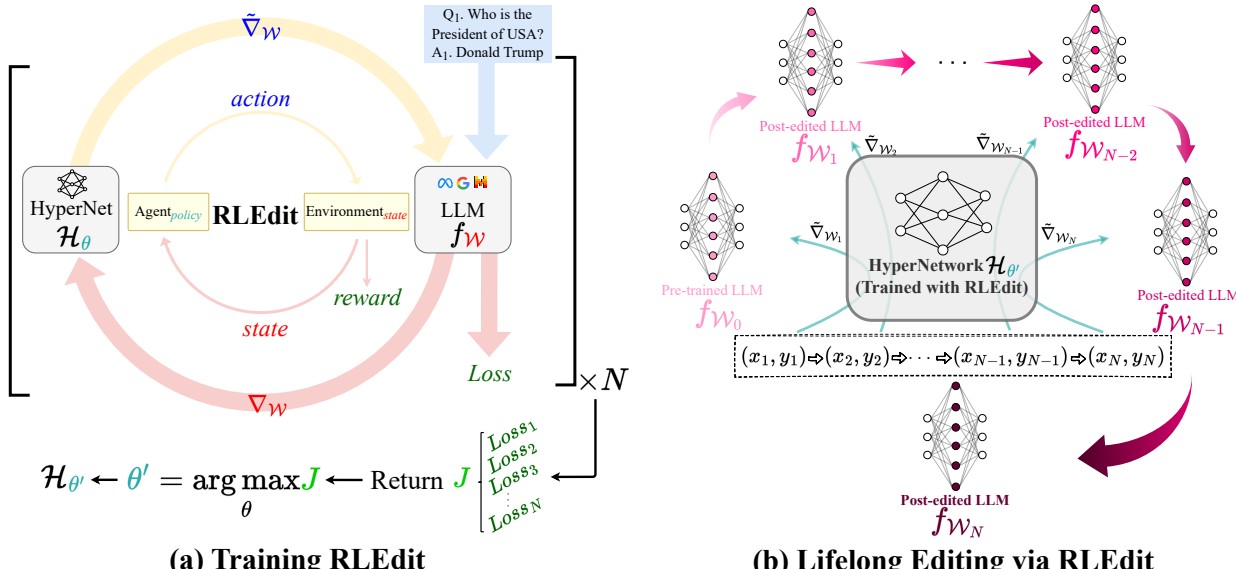

**(a) Training RLEdit**

**(b) Lifelong Editing via RLEdit**

Figure 2: Overview of lifelong editing with RLEdit. (a) illustrates the training process of RLEdit's hypernetwork, while (b) demonstrates how the trained hypernetwork performs lifelong editing. Best viewed in color.

Meanwhile, the reward function $R$ computes the reward $r_i$ for this interaction:

$$R(s_t, a_t) = r_t. \tag{6}$$

This process is illustrated in Figure 2(a). Through this modeling approach, we have successfully formulated the training process of hypernetworks in lifelong editing as an MDP. For more detailed proof, please refer to Appendix D.

### 3.1.2. DESIGN OF THE REWARD FUNCTION

Next, we focus on the design of the reward function $R$ mentioned above, which is crucial for RLEdit to achieve both efficient and effective editing. First, let's recall the fundamental objective of RL: to determine an optimal policy $\pi$ for the agent, where $\pi$ specifies which action $a \sim \pi(s)$ the agent should execute in state $s$ (Sutton & Barto, 1998). The reward function serves as a quantitative measure of policy performance. Specifically, the optimization objective of an RL system is to maximize the total reward $J$ over the entire trajectory. In this work, to align the RL optimization objective with the lifelong editing objective, we design a multi-component reward function for RLEdit as follows.

**Basic Component.** We first incorporate two fundamental objectives of model editing as the basic components of the reward function: target knowledge updating and unrelated knowledge preservation. Formally:

$$\begin{aligned} \mathcal{L}_e &= -\log p_{\mathcal{W}}\left(y_e \mid x_e\right), \\ \mathcal{L}_{loc} &= \mathrm{KL}\left[p_{\mathcal{W}_0}(\cdot|x_{loc}) \,\|\, p_{\mathcal{W}}(\cdot|x_{loc})\right], \end{aligned} \tag{7}$$

where $\mathcal{L}_e$ and $\mathcal{L}_{loc}$ measure the effectiveness of target knowledge updating and unrelated knowledge preservation, respectively; $(x_e, y_e)$ represents the equivalence neighborhood of target knowledge, and $x_{loc}$ represents unrelated

knowledge, both derived from the previously mentioned input knowledge sample $(x, y)$; $\mathcal{W}_0$ denotes the pre-trained parameters of the LLM. Following prior hypernetwork-based editing methods (Mitchell et al., 2022a; Tan et al., 2024; Zhang et al., 2024b), we introduce a coefficient $\lambda_{loc}$ to balance the trade-off between these two terms:

$$\mathcal{L}_{base} = \mathcal{L}_e + \lambda_{loc}\mathcal{L}_{loc}. \tag{8}$$

**Memory Backtracking Component.** In lifelong editing, subsequent edits may interfere with previous ones, potentially degrading the effectiveness of earlier modifications. To address this issue and avoid focusing solely on current knowledge while ignoring previously edited knowledge, we propose a memory backtracking component $\mathcal{L}_{back}$. Specifically, at time step $t$, we not only compute the basic loss of current knowledge $(x_t, y_t)$ on the current LLM $f_{\mathcal{W}_{t-1}}$ according to Equation 8, but also calculate the loss of previous $k$ knowledge samples $(x_i, y_i)_{i=t-k}^{t-1}$ on $f_{\mathcal{W}_{t-1}}$ as the backtracking term using the same equation. In essence, at each time step, we simultaneously consider the basic losses of both current knowledge and the previous $k$ pieces of knowledge on the LLM. Considering that previous knowledge has already been edited and only needs review rather than re-editing, we introduce a decay factor $\mu$, which assigns weights to the losses from previous knowledge through exponential decay based on temporal distance:

$$\mathcal{L}_{back_t} = \sum_{i=t-k}^{t-1} \mu^{t-i}\left(\mathcal{L}_{e_i, \mathcal{W}_{t-1}} + \lambda_{loc}\mathcal{L}_{loc_i, \mathcal{W}_{t-1}}\right), \tag{9}$$

where $\mathcal{L}_{e_i, \mathcal{W}_{t-1}}$ and $\mathcal{L}_{loc_i, \mathcal{W}_{t-1}}$ denotes the loss calculated for the $i$-th batch of knowledge, evaluated using the LLM parameters $\mathcal{W}_{t-1}$ from time step $t-1$.

Ablation studies in Section 4 demonstrate that the generalization improvement from the memory backtracking component is one of the key factors enabling RLEdit to maintain effectiveness through more than 10,000 edits.

**Regularization Component.** Finally, to constrain the magnitude of updates $\tilde{\nabla}_{\mathcal{W}}$ generated by the hypernetwork at each time step, we introduce the $\ell_2$ norm $\|\tilde{\nabla}_{\mathcal{W}}\|^2$ of $\tilde{\nabla}_{\mathcal{W}}$ as a regularization term. The ablation study in Section 4 shows two benefits of this regularization: First, limiting the magnitude of $\tilde{\nabla}_{\mathcal{W}}$ minimizes the disruption to the original parameter distribution, thereby preserving the LLM's general capabilities; Second, it enhances training stability, ensuring the convergence of the hypernetwork.

Based on the above three components, the reward $r_t$ at time step $t$ is formulated as:

$$r_t = -(\mathcal{L}_{base_t} + \mathcal{L}_{back_t} + \eta\|\tilde{\nabla}_{W_t}\|^2), \qquad (10)$$

where $\eta$ serves as the regularization coefficient. With this reward function, the hypernetwork-based lifelong editing task has been completely formulated as an RL problem.

### 3.2. Training Process of RLEdit

The most straightforward training approach would be to optimize the policy (*i.e.*, hypernetwork parameters $\theta$) using gradient ascent after obtaining rewards via Equation 10 at each time step. This online reinforcement learning algorithm, however, becomes computationally expensive when applied at each time step, given that training sets typically contain substantial knowledge to ensure generalization. Moreover, this approach may cause the hypernetwork to overfit specific knowledge samples, thereby reducing its effectiveness in generating general parameter updates. We therefore optimize collectively after traversing the dataset and collecting all rewards along the trajectory. More formally:

$$\theta' = \arg\max_{\theta} J, \qquad (11)$$

$$J = \sum_{i=1}^{n} \gamma^i r_i = -\sum_{i=1}^{n} \gamma^i (\mathcal{L}_{base_i} + \mathcal{L}_{back_i} + \eta\|\tilde{\nabla}_{W_i}\|^2), \quad (12)$$

where $J$ denotes the total reward over the entire trajectory, $\theta'$ represents the optimized hypernetwork parameters, and $\gamma$ represents the discount factor. Here, we set $\gamma = 1$ to ensure that the importance of all knowledge entries across the entire sequence is uniformly weighted during the training phase.

This training approach significantly accelerates hypernetwork convergence. We experimentally demonstrate that RLEdit achieves a 100-fold efficiency improvement over current lifelong editing methods primarily due to this strategy. After updates of the hypernetwork using $J$ in Equation 12, it gradually learns how to edit LLMs with varying parameter states. Moreover, after capturing the intrinsic relationships between knowledge samples, our offline update

---

**Algorithm 1** RLEdit Hypernetwork Training

**Input:** Pre-trained LLM $f_{\mathcal{W}_0}$, hypernetwork $\mathcal{H}$ with initial parameter $\theta$, hyperparameters $k$, $\gamma$, $\lambda_{loc}$ and $\eta$
**Output:** Optimized hypernetwork parameter $\theta'$
**repeat**
    Randomly sample $(x_i, y_i, x_{e_i}, y_{e_i}, x_{loc_i})_{i=1}^n$
    **for** $t = 1$ **to** $n$ **do**
        $\mathcal{L}_t \leftarrow -\log p_{\mathcal{W}_{t-1}}(y_t | x_t)$
        Back-propagate $\mathcal{L}_t$ and cache $\nabla_{\mathcal{W}_{t-1}}$
        $\tilde{\nabla}_{\mathcal{W}_t} \leftarrow \mathcal{H}(\nabla_{\mathcal{W}_{t-1}})$
        $\mathcal{W}_t \leftarrow \mathcal{W}_{t-1} + \tilde{\nabla}_{\mathcal{W}_t}$
        **for** $i = t - k$ **to** $t$ **do**
            $\mathcal{L}_{e_i} \leftarrow -\log p_{\mathcal{W}_t}(y_{e_i} | x_{e_i})$
            $\mathcal{L}_{loc_i} \leftarrow \text{KL}(p_{\mathcal{W}_0}(\cdot | x_{loc_i}) \| p_{\mathcal{W}_t}(\cdot | x_{loc_i}))$
        **end for**
        $\mathcal{L}_t \leftarrow \sum_{i=t-k}^t (\mathcal{L}_{e_i} + \lambda_{loc}\mathcal{L}_{loc_i})$
        $r_t \leftarrow -(\mathcal{L}_t + \eta\|\tilde{\nabla}_{\mathcal{W}_t}\|^2)$
    **end for**
    $J \leftarrow \sum_{t=1}^n \gamma^t r_t$
    Back-propagate $J$ and update $\theta$
**until** hypernetwork convergence
**return** $\theta'$

---

approach enables policy optimization from a higher-level sequential perspective. Ablation studies in Section 4 indicate that this update approach substantially enhances the hypernetwork's effectiveness in lifelong editing.

In summary, the training of RLEdit hypernetwork follows this process. At each time step $t$: (1) We collect gradients $\nabla_{\mathcal{W}_{t-1}}$ from new knowledge sample $(x_t, y_t)$ on LLM $f_{\mathcal{W}_{t-1}}$ through one step of parameter-frozen fine-tuning; (2) Through hypernetwork $\mathcal{H}$, we map $\nabla_{\mathcal{W}_{t-1}}$ to LLM parameter updates $\tilde{\nabla}_{\mathcal{W}_t}$, adding it to $\mathcal{W}_{t-1}$ to obtain $\mathcal{W}_t$; (3) Calculate reward $r_t$ on $f_{\mathcal{W}_t}$ using Equation 10; (4) Repeat steps (1)-(3) until traversing the entire training set, obtaining total reward $J$ via Equation 12; (5) Optimize hypernetwork parameters $\theta$ using $J$ through stochastic gradient descent; (6) Repeat steps (1)-(5) until hypernetwork convergence. This training process yields the final hypernetwork parameters $\theta'$ that demonstrate strong adaptability to lifelong editing tasks. The pseudo-code is provided in Algorithm 1.

### 3.3. Applying RLEdit for Lifelong Editing

After completing the training process in Section 3.2, given knowledge $(x, y)$ to be edited, RLEdit only needs to perform parameter-frozen fine-tuning on LLM $f_{\mathcal{W}}$ to collect gradients $\nabla_{\mathcal{W}}$, feed them into the hypernetwork to generate $\tilde{\nabla}_{\mathcal{W}}$, and add it to $\mathcal{W}$, as shown in Figure 2(b).

In this process, the hypernetwork can typically be constructed with a simple 4-layer MLP, yet effectively handles lifelong editing tasks with varying numbers of knowledge samples. We attribute this excellent performance to the

Table 2: Comparison of RLEdit and baseline methods on lifelong editing tasks. The upper section represents fine-tuning and locate-then-edit methods, while the lower represents hypernetwork-based methods. MEND* and MALMEN* represent methods where the hypernetworks of MEND and MALMEN are retrained for each new knowledge editing.

| Methods | FEVER | | | | | | | | | |
|---|---|---|---|---|---|---|---|---|---|---|
| | LLaMA-3-8B | | | Gemma-2-9B | | | Mistral-7B-v0.3 | | | Time↓ |
| | Eff. | Gen. | Spe. | Eff. | Gen. | Spe. | Eff. | Gen. | Spe. | |
| FT | $1.80_{\pm0.10}$ | $6.39_{\pm0.17}$ | $26.33_{\pm0.13}$ | $34.23_{\pm0.31}$ | $29.22_{\pm0.30}$ | $34.23_{\pm0.31}$ | $17.08_{\pm0.24}$ | $21.57_{\pm0.30}$ | $37.47_{\pm0.32}$ | 0.3358s |
| ROME | $38.56_{\pm0.28}$ | $44.53_{\pm0.29}$ | $9.29_{\pm0.17}$ | $30.07_{\pm0.26}$ | $25.13_{\pm0.11}$ | $10.79_{\pm0.24}$ | $0.00_{\pm0.00}$ | $0.00_{\pm0.00}$ | $0.00_{\pm0.00}$ | 6.0243s |
| MEMIT | $0.18_{\pm0.02}$ | $0.01_{\pm0.01}$ | $0.00_{\pm0.00}$ | $11.12_{\pm0.26}$ | $10.09_{\pm0.25}$ | $8.04_{\pm0.21}$ | $0.00_{\pm0.00}$ | $0.00_{\pm0.00}$ | $0.00_{\pm0.00}$ | 6.3423s |
| PRUNE | $56.64_{\pm0.24}$ | $43.31_{\pm0.18}$ | $0.85_{\pm0.05}$ | $13.34_{\pm0.25}$ | $11.17_{\pm0.27}$ | $9.43_{\pm0.17}$ | $5.27_{\pm0.34}$ | $3.11_{\pm0.16}$ | $5.89_{\pm0.23}$ | 6.3356s |
| RECT | $60.95_{\pm0.27}$ | $52.40_{\pm0.26}$ | $1.75_{\pm0.07}$ | $59.81_{\pm0.23}$ | $54.89_{\pm0.15}$ | $0.05_{\pm0.01}$ | $0.55_{\pm0.04}$ | $0.05_{\pm0.01}$ | $0.00_{\pm0.00}$ | 6.0486s |
| AlphaEdit | $\underline{94.22}_{\pm0.25}$ | **$94.14_{\pm0.18}$** | $25.57_{\pm0.14}$ | $94.37_{\pm0.26}$ | $88.34_{\pm0.13}$ | $31.21_{\pm0.54}$ | $\underline{32.74}_{\pm0.42}$ | $\underline{30.03}_{\pm0.29}$ | $8.44_{\pm0.45}$ | 6.2307s |
| MEND | $0.00_{\pm0.00}$ | $0.00_{\pm0.00}$ | $0.00_{\pm0.00}$ | $50.93_{\pm0.43}$ | $50.76_{\pm0.27}$ | $0.36_{\pm0.04}$ | $0.00_{\pm0.00}$ | $0.00_{\pm0.00}$ | $0.00_{\pm0.00}$ | 0.9175s |
| MEND* | $10.37_{\pm0.19}$ | $9.34_{\pm0.22}$ | $4.88_{\pm0.24}$ | $50.87_{\pm0.24}$ | $52.01_{\pm0.11}$ | $0.39_{\pm0.04}$ | $0.00_{\pm0.00}$ | $0.00_{\pm0.00}$ | $0.00_{\pm0.00}$ | 6.1280s |
| MALMEN | $0.01_{\pm0.01}$ | $0.01_{\pm0.01}$ | $0.14_{\pm0.03}$ | $94.56_{\pm0.09}$ | $91.94_{\pm0.16}$ | $68.65_{\pm0.33}$ | $16.67_{\pm0.21}$ | $16.61_{\pm0.18}$ | $12.16_{\pm0.16}$ | 1.9858s |
| MALMEN* | $5.74_{\pm0.20}$ | $5.29_{\pm0.13}$ | $1.26_{\pm0.18}$ | $88.32_{\pm0.33}$ | $84.29_{\pm0.33}$ | **$69.70_{\pm0.27}$** | $17.73_{\pm0.24}$ | $13.30_{\pm0.27}$ | $13.85_{\pm0.21}$ | 9.3358s |
| DAFNet | $31.27_{\pm0.47}$ | $28.82_{\pm0.43}$ | $\underline{66.55}_{\pm0.41}$ | $20.78_{\pm0.31}$ | $19.99_{\pm0.35}$ | $53.10_{\pm0.56}$ | $4.86_{\pm0.14}$ | $4.21_{\pm0.19}$ | $\underline{41.71}_{\pm0.48}$ | 8.2553s |
| **RLEdit** | **$95.34_{\pm0.34}$** | $\underline{93.58}_{\pm0.38}$ | **$70.36_{\pm0.29}$** | **$95.44_{\pm0.21}$** | **$92.83_{\pm0.30}$** | $68.76_{\pm0.18}$ | **$88.99_{\pm0.22}$** | **$88.25_{\pm0.25}$** | **$73.64_{\pm0.11}$** | **0.2238s** |

| Methods | ZsRE | | | | | | | | | |
|---|---|---|---|---|---|---|---|---|---|---|
| | LLaMA-3-8B | | | Gemma-2-9B | | | Mistral-7B-v0.3 | | | Time↓ |
| | Eff. | Gen. | Spe. | Eff. | Gen. | Spe. | Eff. | Gen. | Spe. | |
| FT | $17.10_{\pm0.22}$ | $16.73_{\pm0.22}$ | $8.27_{\pm0.13}$ | $12.90_{\pm0.20}$ | $13.09_{\pm0.20}$ | $0.07_{\pm0.02}$ | $32.84_{\pm0.30}$ | $33.78_{\pm0.30}$ | **$42.19_{\pm0.31}$** | 0.3366s |
| ROME | $0.54_{\pm0.04}$ | $0.57_{\pm0.04}$ | $0.40_{\pm0.02}$ | $3.45_{\pm0.32}$ | $3.33_{\pm0.11}$ | $8.24_{\pm0.29}$ | $0.00_{\pm0.00}$ | $0.00_{\pm0.00}$ | $0.00_{\pm0.00}$ | 6.0012s |
| MEMIT | $0.00_{\pm0.00}$ | $0.00_{\pm0.00}$ | $0.13_{\pm0.02}$ | $5.23_{\pm0.21}$ | $3.11_{\pm0.12}$ | $5.98_{\pm0.12}$ | $0.00_{\pm0.00}$ | $0.00_{\pm0.00}$ | $0.13_{\pm0.02}$ | 6.0677s |
| PRUNE | $12.27_{\pm0.43}$ | $12.01_{\pm0.23}$ | $9.88_{\pm0.29}$ | $10.21_{\pm0.27}$ | $8.88_{\pm0.29}$ | $11.95_{\pm0.55}$ | $0.00_{\pm0.00}$ | $0.00_{\pm0.00}$ | $0.00_{\pm0.00}$ | 6.1588s |
| RECT | $11.05_{\pm0.41}$ | $8.12_{\pm0.15}$ | $28.12_{\pm0.13}$ | $12.45_{\pm0.45}$ | $10.11_{\pm0.51}$ | $26.09_{\pm0.44}$ | $8.18_{\pm0.33}$ | $8.04_{\pm0.46}$ | $11.32_{\pm0.49}$ | 6.6558s |
| AlphaEdit | $\underline{86.83}_{\pm0.23}$ | $\underline{81.48}_{\pm0.28}$ | $29.09_{\pm0.22}$ | $81.18_{\pm0.33}$ | $73.24_{\pm0.46}$ | $30.34_{\pm0.19}$ | $0.00_{\pm0.00}$ | $0.00_{\pm0.00}$ | $0.00_{\pm0.00}$ | 6.1831s |
| MEND | $0.00_{\pm0.00}$ | $0.00_{\pm0.00}$ | $0.00_{\pm0.00}$ | $0.00_{\pm0.00}$ | $0.00_{\pm0.00}$ | $0.00_{\pm0.00}$ | $0.00_{\pm0.00}$ | $0.00_{\pm0.00}$ | $0.00_{\pm0.00}$ | 0.9686s |
| MEND* | $0.00_{\pm0.00}$ | $0.00_{\pm0.00}$ | $0.00_{\pm0.00}$ | $8.84_{\pm0.24}$ | $8.45_{\pm0.21}$ | $10.21_{\pm0.19}$ | $0.00_{\pm0.00}$ | $0.00_{\pm0.00}$ | $0.00_{\pm0.00}$ | 5.9265s |
| MALMEN | $9.87_{\pm0.12}$ | $9.00_{\pm0.09}$ | $2.11_{\pm0.15}$ | $46.60_{\pm0.33}$ | $42.50_{\pm0.32}$ | $19.66_{\pm0.35}$ | $0.00_{\pm0.00}$ | $0.00_{\pm0.00}$ | $0.00_{\pm0.00}$ | 2.2779s |
| MALMEN* | $12.23_{\pm0.11}$ | $11.08_{\pm0.22}$ | $2.43_{\pm0.09}$ | $56.97_{\pm0.24}$ | $44.28_{\pm0.29}$ | $15.02_{\pm0.21}$ | $0.01_{\pm0.01}$ | $0.02_{\pm0.01}$ | $1.25_{\pm0.04}$ | 9.4277s |
| DAFNet | $21.99_{\pm0.47}$ | $11.17_{\pm0.43}$ | $\underline{32.21}_{\pm0.39}$ | $5.94_{\pm0.24}$ | $5.68_{\pm0.33}$ | $\underline{36.29}_{\pm0.45}$ | $1.25_{\pm0.08}$ | $2.12_{\pm0.12}$ | $25.27_{\pm0.54}$ | 8.2383s |
| **RLEdit** | **$89.42_{\pm0.34}$** | **$87.32_{\pm0.23}$** | **$44.78_{\pm0.50}$** | **$84.37_{\pm0.22}$** | **$79.82_{\pm0.26}$** | **$37.15_{\pm0.41}$** | **$71.12_{\pm0.31}$** | **$67.42_{\pm0.27}$** | $\underline{27.43}_{\pm0.44}$ | **0.2224s** |

enhanced generalization capability brought by RL.

**Boosting Current Methods with RLEdit.** Since RLEdit's superior efficiency and effectiveness primarily stem from its RL paradigm, it can serve as a plug-and-play module to integrate with most existing hypernetwork-based approaches. Specifically, hypernetwork-based methods consist of three fundamental elements: (1) hypernetwork architecture, (2) selection of hypernetwork inputs, and (3) loss function for training the hypernetwork. For the first two elements, we can directly adopt their corresponding components from RLEdit. For the third element, we can use it to replace the basic term in RLEdit's reward function (Equation 10). Our experimental results in Section 4.6 demonstrate these optimizations, further validating the benefits of formulating hypernetwork training in lifelong editing as an RL paradigm.

## 4. Experiments

We conduct extensive experiments to evaluate both the effectiveness and efficiency of our approach. Additionally, we perform ablation studies to analyze the contribution of each component in RLEdit, which can be found in Appendix B.1.

### 4.1. Experimental Settings

We begin with a brief overview of the LLMs, datasets, evaluation metrics, and baseline methods used in our experiments. More detailed information can be found in Appendix A.

**Base LLMs.** We conduct experiments on three 8B-scale autoregressive LLMs: Llama-3-8B[1], Gemma-2-9B (Team et al., 2024), and Mistral-7B-v0.3 (Jiang et al., 2023).

**Datasets & Evaluation Metrics.** We evaluate RLEdit on three widely-used datasets: ZsRE (Levy et al., 2017), FEVER (Thorne et al., 2018), and CounterFact (Meng et al., 2022). Following previous evaluation standards (Mitchell et al., 2022a; Meng et al., 2022; 2023), we use three metrics to assess editing success for each dataset: Efficacy, Generalization, and Specificity, as described in Appendix A.3.

**Baseline Methods.** We compare RLEdit against multiple editing methods, including FT (Zhu et al., 2021), ROME (Meng et al., 2022), MEND (Mitchell et al., 2022a), MALMEN (Tan et al., 2024), DAFNet (Zhang et al., 2024b), MEMIT (Meng et al., 2023), PRUNE (Ma et al., 2024), RECT (Gu et al., 2024), and AlphaEdit (Fang et al., 2024).

---

[1]https://llama.meta.com/llama3

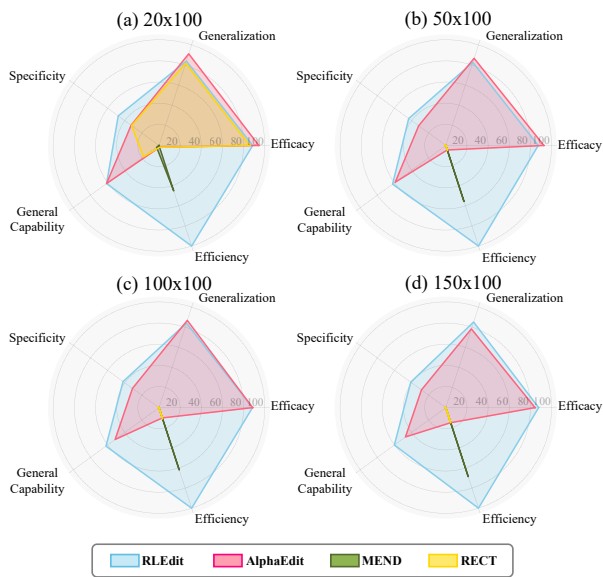

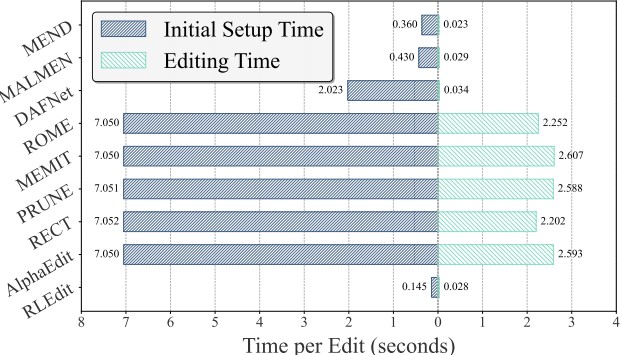

Figure 4: Per-sample editing time comparison across methods. Best viewed in color.

Figure 3: Performance comparison of editing methods across multiple metrics: Efficacy, Generalization, Specificity, General Capability, and Efficiency. (a)-(d) show results for 20×100, 50×100, 100×100, and 150×100 configurations. For General Capability, we measure that using the average of six GLUE metrics in Section 4.5; for Efficiency, we calculate the ratio between the total editing time of each method and that of the fastest method. Best viewed in color.

### 4.2. Performance on Knowledge Update and Preservation

To evaluate the performance on both updating of target knowledge and preservation of unrelated information in lifelong editing tasks, we conducted sequential editing experiments comparing RLEdit against baseline methods on three LLMs, measuring performance across all metrics. We randomly sampled 8,000 knowledge samples from ZsRE and FEVER respectively, performing edits over 400 batches with 20 knowledge samples per batch (denoted as a 400×20 configuration throughout this paper). Upon completion of all batch edits, we evaluate all knowledge metrics on the post-edited LLM. Results are presented in Table 2. The "Time" column in Table 2 indicates the average editing time per knowledge sample, including the time spent on covariance matrices computing or hypernetwork training.

In Table 2, conventional single-step methods (*e.g.*, ROME, MEND, MEMIT) demonstrate poor performance, with Efficacy and Generalization metrics falling below 50%, revealing their inability to handle knowledge conflicts and forgetting issues in lifelong editing scenarios. Other sequential editing methods (*e.g.*, PRUNE, RECT, DAFNet) show varying limitations across datasets, failing to maintain satisfactory performance across all metrics. In contrast, RLEdit achieves superior results across all tested LLMs and datasets, maintaining strong performance in all three metrics. Specif-

ically, RLEdit demonstrates average improvements of 66% in Efficacy, 65% in Generalization, and 40% in Specificity compared to baseline methods, while requiring only 4% of the time compared to most of them.

### 4.3. Editing with Varying Numbers of Knowledge

To verify RLEdit's versatility across different lifelong editing scenarios, we investigated its performance under multiple configurations. We conducted extensive experiments on Llama-3-8B using 20×100, 50×100, 100×100, and 150×100 configurations on ZsRE dataset. The results are presented in Figure 3. We also performed experiments to investigate the impact of varying knowledge batch sizes on performance. Detailed results can be found in Appendix B.

Figure 3 shows that as the number of edits increases, baseline methods show progressive performance degradation, with catastrophic forgetting occurring beyond certain thresholds. RLEdit, however, consistently demonstrates superior performance across all configurations, outperforming baseline methods on most evaluation metrics. RLEdit represents the first method to successfully scale lifelong editing beyond 10,000 knowledge samples. Unlike existing approaches, RLEdit maintains stable performance as knowledge quantity increases, suggesting potential applicability to even longer sequences. These results demonstrate RLEdit's effectiveness for long-term lifelong editing tasks.

### 4.4. Editing Efficiency

To evaluate RLEdit's efficiency, we compared the average editing time per knowledge sample across most methods. Tests were conducted on 3 LLMs using ZsRE dataset under 20×100 configuration while keeping all other variables constant (such as the number of LLM editing layers) to ensure fair comparison. For locate-then-edit methods, the time included covariance matrix computation, parameter update calculation, and edit execution. For hypernetwork-based methods, we measured hypernetwork training, parameter update calculation, and edit execution time. Since covari-

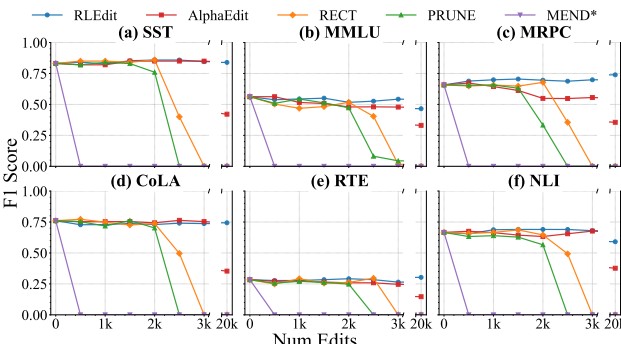

Figure 5: LLM capability assessment using F1 scores on six GLUE tasks. (a)-(f) shows results for SST, MMLU, MRPC, CoLA, RTE, and NLI respectively. Best viewed in color.

ance matrices and trained hypernetworks can be reused, we also analyzed editing time excluding these initial setup costs. Results are presented in Figure 4.

Hypernetwork-based methods demonstrate superior speed compared to locate-then-edit approaches, primarily due to better generalization capabilities. Locate-then-edit methods require complex matrix operations for each edit, resulting in slower execution. RLEdit requires only 0.145 seconds per sample for training, outperforming other hypernetwork-based methods like MEND, MALMEN, and DAFNet. This efficiency derives from RLEdit's sequence-based approach for training, enabling better generalization with reduced training data. RLEdit achieves superior editing results while requiring only 1.14% of the time needed by locate-then-edit methods, demonstrating its exceptional efficiency and promising potential in long-term lifelong editing scenarios.

### 4.5. General Capability Tests

To evaluate how lifelong editing affects LLMs' general capabilities, we assessed downstream performance using 6 tests from GLUE (Wang et al., 2019): SST (Socher et al., 2013), MMLU (Hendrycks et al., 2021), MRPC (Dolan & Brockett, 2005), CoLA (Warstadt et al., 2019), RTE (Bentivogli et al., 2009), and NLI (Williams et al., 2018). We measured F1 Scores on Llama-3-8B using ZsRE dataset under $30 \times 100$ configuration, and the results are shown in Figure 5. Complete tests are available in Appendix B.

Baseline methods show progressive degradation of general capabilities as edited knowledge samples increase, with some methods leading to complete performance collapse. This degradation likely stems from cumulative editing errors, caused by causal trace instabilities or imprecise hypernetwork training. In contrast, RLEdit maintains consistent performance across all tests, with LLMs edited for 3,000 knowledge samples performing comparable to their pre-trained versions. This stability, achieved through regularization, demonstrates RLEdit's ability to both effectively edit target knowledge and preserve the model's general capabilities.

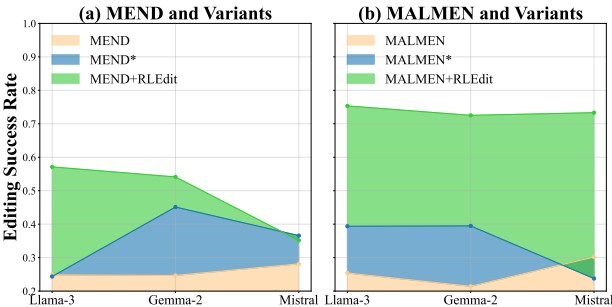

Figure 6: Performance comparison of hypernetwork-based methods (MEND and MALMEN) with and without RLEdit framework integration, including variants with hypernetwork retraining before each edit. Results presented in the figure show the weighted average of metrics across CounterFact and ZsRE, where Efficacy and Generalization are assigned weights of 1, and Specificity is assigned a weight of 0.5. Best viewed in color.

### 4.6. Improving MEND/MALMEN

While existing hypernetwork-based editing methods focus on single-edit scenarios, RLEdit's RL framework and training methodology specifically address lifelong editing challenges, which could integrate with other hypernetwork-based methods as a plug-and-play module. To demonstrate this portability, we conducted $20 \times 100$ editing experiments across 3 LLMs and 3 datasets, comparing MEND, MEND*, MEND+RLEdit, MALMEN, MALMEN*, and MALMEN+RLEdit. Results are shown in Figure 6.

The results demonstrate RLEdit's portability across several methods. After incorporating RLEdit's training methodology, MEND and MALMEN showed average improvements of 22.89% and 51.86% respectively across all metrics, despite their initial suboptimal performance in lifelong editing tasks. This confirms RLEdit's effectiveness as a plug-and-play module for enabling lifelong editing capabilities in both existing and future hypernetwork-based methods.

## 5. Related Works

**Parameter-modifying Editing.** Parameter-modifying methods modify LLM parameters through various approaches to achieve knowledge editing. One approach is locate-then-edit method. Knowledge-neuron (Dai et al., 2022) pioneered the use of causal analysis to locate knowledge in LLMs. ROME (Meng et al., 2022) adopted this idea, using causal trace to identify factual associations and edit knowledge in MLP layers. MEMIT (Meng et al., 2023) extended this approach to mass-editing. PMET (Li et al., 2024) specifically investigated the role of MHSA in model editing. ECE (Zhang et al., 2025c) incorporates LLM explainability into the editing process and clusters similar knowledge based on explanation results. Another approach is meta-learning. KE (De Cao

et al., 2021) proposed using hypernetwork for knowledge editing. MEND (Mitchell et al., 2022a) applied low-rank decomposition to LLM fine-tuning gradients, enabling faster and more effective training in hypernetworks. MALMEN (Tan et al., 2024) aggregated parameter shifts using normal equations, extending hypernetworks to mass-editing.

**Parameter-preserving Editing.** Parameter-preserving methods guide LLM outputs by incorporating external modules or additional parameters. SERAC (Mitchell et al., 2022b) stores edits in additional memory and uses classifiers for edit matching. T-Patcher (Huang et al., 2023) edits knowledge by using additional neurons. IKE (Zheng et al., 2023) improves LLM's in-context learning capabilities using memory retrieval-based examples. OneEdit (Zhang et al., 2024a) proposed a neural-symbolic prototype system for collaborative knowledge editing using natural language.

**Lifelong Model Editing.** To meet the requirements of lifelong model editing, researchers have studied sequential editing methods. T-Patcher (Huang et al., 2023) achieves sequential editing by continuously introducing additional correction neurons. GRACE (Hartvigsen et al., 2023) enables sequential knowledge editing through dynamic updates of external codebook modules. WISE (Wang et al., 2024) uses a knowledge-sharding mechanism to optimize the balance of various metrics in sequential editing. RECT (Gu et al., 2024) studies the impact of regularization on sequential editing, using smaller parameter shifts to maintain stability. PRUNE (Ma et al., 2024) reduces disturbance to original knowledge by limiting singular values of the edit update matrix. O-Edit (Cai & Cao, 2024) orthogonalizes knowledge update directions to reduce mutual interference. AlphaEdit (Fang et al., 2024) maintains knowledge stability by projecting existing knowledge into null space. NSE (Jiang et al., 2024) employs neuron-level sorting to selectively identify and update influential neurons, while utilizing weights rewinding to prevent model failures during sequential editing.

## 6. Conclusion

In this work, we represent RLEdit, a hypernetwork-based editing method designed for lifelong editing. RLEdit formulates lifelong editing as an RL task, employing an offline update approach to enhance the model's retention of entire knowledge sequences. Additionally, RLEdit proposes the use of memory backtracking to review previously edited knowledge and applies regularization to mitigate knowledge forgetting over long sequences. Through extensive testing on several LLMs across multiple datasets, our experimental results demonstrate that RLEdit significantly outperforms existing baseline methods in lifelong editing tasks, showing superior performance in editing effectiveness, editing efficiency, and general capability preservation.

## Limitations

While RLEdit demonstrates promising results in lifelong editing tasks, several limitations should be acknowledged. Our evaluation methodology follows conventional datasets from existing model editing literature, primarily focusing on factual knowledge modifications without exploring other data domains. Furthermore, the capability of post-edited LLMs in processing multi-hop information remains unexplored in our current study. Although achieving robust lifelong editing capabilities continues to pose significant challenges, our future work will extend these experiments, potentially providing valuable insights for advancing research in lifelong editing.

## Impact Statement

RLEdit significantly enhances the capabilities of lifelong model editing, making it invaluable for updating and maintaining knowledge in real-world applications. Given that the ability to directly modify model parameters introduces potential risks, such as the injection of false or harmful information, we strongly urge researchers to implement strict validation and oversight to ensure the ethical use of these techniques. The original goal of our work is positive, aiming to facilitate efficient knowledge updates in large language models. We encourage researchers to leverage this technology responsibly while maintaining appropriate safeguards for its deployment.

## Acknowledgement

This research is supported by the National Science and Technology Major Project (2023ZD0121102), the Beijing Natural Science Foundation (4252023), and the Innovation Funding of ICT, CAS (E361120).

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

# A. Detailed Experimental Setup

In this section, we elaborate on our experimental setup, which consists of five parts: baseline methods, datasets, evaluation metrics, GLUE benchmarks, and hyperparameter configuration. Most experiments were conducted on a single NVIDIA A100 (80GB) GPU. Editing time for all methods was measured using LLMs in half-precision mode. To better simulate real-world applications, we used the instruction tuning versions of LLMs.

## A.1. Baseline Methods

We utilized the code from AlphaEdit and MALMEN to evaluate the performance of baseline methods. The baseline methods used in this paper are as follows:

- **FT-L** (Zhu et al., 2021) is a knowledge editing approach that focuses on fine-tuning specific layers of the LLM through autoregressive loss. We implemented this baseline method using the hyperparameter settings from the original paper.

- **MEND** (Mitchell et al., 2022a) is an efficient editing method based on hypernetworks. It trains a hypernetwork to learn patterns in knowledge editing by mapping low-rank decomposed fine-tuning gradients to LLM parameter updates. This approach enables efficient and localized knowledge editing. We implemented this baseline method using the hyperparameter settings from the original paper, completing training over the entire training set. Additionally, we introduce MEND* as a baseline. To address the mismatch between the initial hypernetwork and post-edited LLM in lifelong editing scenarios, we periodically retrain the hypernetwork using post-edited parameters. We adopted a strategy of retraining the hypernetwork every three editing batches.

- **ROME** (Meng et al., 2022) is a method for updating specific factual associations in LLM parameters. It identifies key neuron activations in MLP layers through perturbation-based knowledge localization, then modifies MLP layer weights by computing Lagrange remainders to edit knowledge. Since ROME doesn't support massive editing, we followed the original paper's configuration and evaluated it through multiple batches of single editing.

- **MEMIT** (Meng et al., 2023) is a method supporting large-batch knowledge updates. Building upon ROME's modeling approach, MEMIT extends it by using least squares approximation to directly manipulate parameters at specific layers, enabling multi-layer updates. This allows MEMIT to simultaneously update hundreds or thousands of knowledge facts. We evaluated MEMIT's performance in lifelong editing using the original configuration from its paper.

- **MALMEN** (Tan et al., 2024) is a hypernetwork-based method designed for massive editing. To aggregate parameter shifts across large batches of knowledge, MALMEN employs a least squares approach, deriving optimal parameter shifts by solving normal equations. This algorithm effectively addresses knowledge conflicts in massive editing scenarios. We implemented this baseline method using the original paper's hyperparameter configuration and completed training across the entire training set. Similar to MEND, we also introduce MALMEN* as a baseline.

- **DAFNet** (Zhang et al., 2024b) is a model editing method specifically designed for sequential editing. It features a dynamic auxiliary fusion network that enhances semantic interactions between knowledge triples in the sequence, enabling continuous mistake rectification. Through this auxiliary network, DAFNet improves the performance of hypernetwork approaches in sequential editing tasks. We implemented this baseline method using the original paper's hyperparameter configuration and completed training across ZsRE and CounterFact datasets.

- **PRUNE** (Ma et al., 2024) is an editing method focused on sequential editing scenarios. By imposing conditional restraints on edited matrices, PRUNE limits the interference of new knowledge on previously stored model knowledge, thereby addressing the problem of model performance decline during multiple sequential edits. We implemented this baseline method using the hyperparameter configuration from their original paper.

- **RECT** (Gu et al., 2024) is an editing method designed to minimize the impact of editing on LLM's general capabilities. It investigates the role of regularization in lifelong editing and prevents editing overfitting by regularizing weight updates during the editing process. This enables RECT to achieve high editing performance while maintaining the LLM's general capabilities. We implemented this baseline method using the hyperparameter configuration from their original paper.

- **AlphaEdit** (Fang et al., 2024) is an editing method aimed at mitigating knowledge disruption in LLM lifelong editing. By introducing the concept of null space, AlphaEdit projects parameter updates onto a knowledge-preserving null space

before applying them, thus reducing interference between different knowledge updates. AlphaEdit has been proven to achieve SOTA performance across multiple metrics while maintaining strong transferability. We implemented this baseline method using the hyperparameter configuration from their original paper.

## A.2. Datasets

Next, we introduce the datasets used in this paper.

- **ZsRE** (Zero-shot Relation Extraction) (Levy et al., 2017) dataset serves as a benchmark dataset in the field of language model editing research. The dataset's structure incorporates three distinct components for each entry: a primary question and its corresponding answer intended for editing purposes, multiple paraphrased variations of the original question created using back-translation techniques, and locality questions that are semantically unrelated to the original query. This comprehensive structure enables researchers to evaluate model editing performance across three critical dimensions: accuracy in incorporating new information, robustness when faced with differently worded but semantically equivalent queries, and precision in maintaining unrelated knowledge without interference. For locate-then-edit methods, we use the version from MEMIT; for hypernetwork-based methods, we use the version from MEND, where ZsRE is divided into training and test sets for hypernetwork training and editing performance evaluation respectively.

- **CounterFact** (Meng et al., 2022) represents an advanced dataset specifically designed to evaluate language models' ability to handle contradictory factual information. The dataset's distinctive feature lies in its use of false statements that require correction, making it particularly challenging since models typically provide incorrect answers before editing. Each entry in the dataset contains three elements: an original false statement requiring editing, semantically equivalent rephrased versions of the statement, and unrelated statements for locality purposes. For locate-then-edit methods, we use the version from MEMIT; for hypernetwork-based methods, we also use the version from MEMIT and divide it into training and test sets, each containing approximately 10,000 knowledge instances.

- **FEVER** (Fact Extraction and VERification) (Thorne et al., 2018) dataset is a comprehensive dataset for fact-checking tasks, constructed through systematic modification of Wikipedia content. The dataset contains claims that were created by altering original Wikipedia sentences and then independently verified without reference to their source material. In its structure, FEVER implements a three-category classification system: claims can be marked as Supported, Refuted, or NotEnoughInfo. For claims classified as either Supported or Refuted, the dataset includes supporting evidence sentences that justify the classification decision. When used in editing tasks, the dataset is often simplified to a binary classification problem, where the editing targets are equally distributed between two possible labels (1 and 0), representing the veracity of the claims. For locate-then-edit methods, we extract the subject from queries to adapt to $(s, r, o)$ modeling; for hypernetwork-based methods, we use the version from MEND, where FEVER is divided into training and test sets.

## A.3. Metrics

### A.3.1. ZsRE METRICS

Following previous research (Meng et al., 2022; Mitchell et al., 2022a), we evaluate various model editing methods using standard metrics on the ZsRE dataset. Specifically, given an LLM $f_{\mathcal{W}}$, an editing knowledge pair $(x, y)$, equivalent knowledge $x_e$, and unrelated knowledge pairs $(x_{loc}, y_{loc})$, we examine the following three metrics:

**Efficacy.** This metric measures the success rate of editing knowledge $x$ in $f_{\mathcal{W}}$. It compares the top-1 logits output $y' = f_{\mathcal{W}}(x)$ with the target output $y$ when inputting $x$ into $f_{\mathcal{W}}$:

$$\mathbb{E}\left\{ y = \arg\max_{y'} \mathbb{P}_{f_{\mathcal{W}}}(y' \,|x) \right\}. \tag{13}$$

**Generalization.** This metric measures the success rate of editing equivalent knowledge $x_e$ in $f_{\mathcal{W}}$. It evaluates whether the LLM has truly learned the intrinsic relationships of the knowledge and can extend to other equivalent knowledge. We compare the top-1 logits output $y' = f_{\mathcal{W}}(x_e)$ with the target output $y$ when inputting $x_e$ into $f_{\mathcal{W}}$:

$$\mathbb{E}\left\{ y = \arg\max_{y'} \mathbb{P}_{f_{\mathcal{W}}}(y' \,|x_e) \right\}. \tag{14}$$

**Specificity.** This metric measures the retention rate of unrelated knowledge $x_{loc}$ after editing, examining whether the knowledge editing maintains locality and only modifies the target knowledge. We compare the top-1 logits output $y' = f_{\mathcal{W}}(x_{loc})$ with the original output $y_{loc}$ when inputting $x_{loc}$ into $f_{\mathcal{W}}$:

$$\mathbb{E}\left\{y_{loc} = \arg\max_{y'} \mathbb{P}_{f_{\mathcal{W}}}(y'|x_{loc})\right\}. \tag{15}$$

### A.3.2. COUNTERFACT METRICS

Similarly, following previous research (Meng et al., 2022; 2023), we evaluate various model editing methods using standard metrics on the CounterFact dataset. We follow ROME's original setup, comparing the probabilities of different answers in the logits for measurement. Specifically, given an LLM $f_{\mathcal{W}}$, an editing knowledge pair $(x, y)$, original knowledge pair $(x, y_0)$, equivalent knowledge $x_e$, and unrelated knowledge pairs $(x_{loc}, y_{loc})$, we examine the following three metrics, calculated following ROME and MEMIT:

**Efficacy.** This metric measures the success rate of editing knowledge $x$ in $f_{\mathcal{W}}$. We compare whether the probability of the target output $y$ is higher than the probability of the original answer $y_0$ in the logits when inputting $x$ into $f_{\mathcal{W}}$:

$$\mathbb{E}\left[\mathbb{P}_{f_{\mathcal{W}}}[y|x] > \mathbb{P}_{f_{\mathcal{W}}}[y_0|x]\right]. \tag{16}$$

**Generalization.** This metric measures the success rate of editing equivalent knowledge $x_e$ in $f_{\mathcal{W}}$. We compare whether the probability of the target output $y$ is higher than the probability of the original answer $y_0$ in the logits when inputting $x_e$ into $f_{\mathcal{W}}$:

$$\mathbb{E}\left[\mathbb{P}_{f_{\mathcal{W}}}[y|x_e] > \mathbb{P}_{f_{\mathcal{W}}}[y_0|x_e]\right]. \tag{17}$$

**Specificity.** This metric measures the retention rate of unrelated knowledge $x_{loc}$ after editing. We compare whether the probability of the original answer $y_{loc}$ is higher than the probability of the edited output $y$ in the logits when inputting $x_{loc}$ into $f_{\mathcal{W}}$:

$$\mathbb{E}\left[\mathbb{P}_{f_{\mathcal{W}}}[y_{loc}|x_{loc}] > \mathbb{P}_{f_{\mathcal{W}}}[y|x_{loc}]\right]. \tag{18}$$

### A.3.3. FEVER METRICS

We also evaluate various model editing methods using standard metrics on the FEVER dataset. Specifically, given an LLM $f_{\mathcal{W}}$, an editing knowledge pair $(x, y)$, equivalent knowledge $x_e$, and unrelated knowledge pairs $(x_{loc}, y_{loc})$, we examine the following three metrics:

**Efficacy.** This metric measures the success rate of editing knowledge $x$ in $f_{\mathcal{W}}$. It compares whether the top-1 logits output $y' = f_{\mathcal{W}}(x)$ matches the target output $y$ when inputting $x$ into $f_{\mathcal{W}}$:

$$\mathbb{E}\left\{y = \arg\max_{y'} \mathbb{P}_{f_{\mathcal{W}}}(y'|x)\right\}. \tag{19}$$

**Generalization.** This metric measures the success rate of editing equivalent knowledge $x_e$ in $f_{\mathcal{W}}$. Since we need to examine whether a knowledge edit is truly successful, we must verify if the LLM has genuinely learned the intrinsic relationships of the knowledge and can extend to other equivalent knowledge. We compare whether the top-1 logits output $y' = f_{\mathcal{W}}(x_e)$ matches the target output $y$ when inputting $x_e$ into $f_{\mathcal{W}}$:

$$\mathbb{E}\left\{y = \arg\max_{y'} \mathbb{P}_{f_{\mathcal{W}}}(y'|x_e)\right\}. \tag{20}$$

**Specificity.** This metric measures the retention rate of unrelated knowledge $x_{loc}$ after editing, examining whether the knowledge editing maintains locality and only modifies the target knowledge. We compare whether the top-1 logits output $y' = f_{\mathcal{W}}(x_{loc})$ matches the original output $y_{loc}$ when inputting $x_{loc}$ into $f_{\mathcal{W}}$:

$$\mathbb{E}\left\{y_{loc} = \arg\max_{y'} \mathbb{P}_{f_{\mathcal{W}}}(y'|x_{loc})\right\}. \tag{21}$$

### A.4. GLUE Benchmark

GLUE (General Language Understanding Evaluation) (Wang et al., 2019) benchmark is a collection of resources for training, evaluating, and analyzing natural language understanding systems. We selected 6 metrics from this benchmark to evaluate how well different methods maintain general language capabilities.

- **Stanford Sentiment Treebank (SST)** (Socher et al., 2013) is a dataset consisting of movie review sentences with their associated sentiment labels. This binary classification task requires models to categorize the sentiment expressed in each individual sentence.

- **Massive Multi-task Language Understanding (MMLU)** (Hendrycks et al., 2021) is a comprehensive benchmark designed to assess language models' capabilities across multiple domains. It specifically evaluates model performance in zero-shot and few-shot learning scenarios.

- **Microsoft Research Paraphrase Corpus (MRPC)** (Dolan & Brockett, 2005) serves as a benchmark for evaluating semantic similarity. The task challenges models to identify whether two given sentences convey the same meaning.

- **Recognizing Textual Entailment (RTE)** (Bentivogli et al., 2009) examines logical relationships between sentences. The task requires determining whether a given premise sentence logically implies a hypothesis sentence.

- **Corpus of Linguistic Acceptability (CoLA)** (Warstadt et al., 2019) focuses on grammatical judgment. This single-sentence classification task uses sentences extracted from linguistics publications, requiring models to distinguish between grammatically acceptable and unacceptable sentences.

- **Natural Language Inference (NLI)** (Williams et al., 2018) evaluates natural language understanding capabilities. The benchmark requires models to analyze sentence pairs and determine their logical relationships.

### A.5. Hyperparameter Configuration

Now we describe the hyperparameter configurations in our experiments.

For the hyperparameters in RLEdit training and editing, we set the memory backtracking decay factor $\mu$ to 0.95, the backtracking depth $k$ to 10, the regularization coefficient $\eta$ to 1e-4 and the discount factor $\gamma$ to 1 in the total reward formula. Additionally, the initial learning rate was set to 1e-6, while the meta-learning rate was set to 1e-5. The specific hyperparameter configurations for different models and datasets are shown in Table 3, where rank refers to the rank of linear transformation in hypernetwork, loc_coef refers to the weight coefficient $\lambda_{loc}$ of the locality loss function $\mathcal{L}_{loc}$. Table 3 represent settings that were empirically selected based on their strong experimental performance. Our analysis revealed that the performance of RLEdit is notably sensitive to the choice of the "Layer" parameter, while exhibiting limited sensitivity to variations in "Rank" and "loc_coef".

Table 3: The specific hyperparameter configurations for different models and datasets.

| Datasets | Models | Layer | Rank | loc_coef $\lambda_{loc}$ |
|---|---|---|---|---|
| ZsRE | Llama-3-8B | gate[11-15], up[18-24] | 1024 | 0.6 |
| | Gemma-2-9B | gate[32-40], up[32-40] | 512 | 0.6 |
| | Mistral-7B | down[17, 18] | 1024 | 0.8 |
| CounterFact | Llama-3-8B | gate[22-30], up[22-30] | 512 | 0.6 |
| | Gemma-2-9B | gate[32-40], up[32-40] | 1024 | 0.6 |
| | Mistral-7B | down[17, 18, 19] | 1024 | 0.8 |
| FEVER | Llama-3-8B | gate[22-30], up[22-30] | 1024 | 0.6 |
| | Gemma-2-9B | gate[32-40], up[32-40] | 1024 | 0.6 |
| | Mistral-7B | down[17, 18] | 1024 | 0.6 |

# B. More Experimental Results

In this section, we present additional experimental results.

## B.1. Ablation Study

To assess the contribution of each component in RLEdit, we conducted ablation studies on 3 models and 3 datasets by removing the RL training framework, memory backtracking, and regularization respectively. Table 4 shows the contribution of each component to RLEdit under the 20×100 task.

Table 4: Ablation Study Results for RLEdit.

| Model | Dataset / Method | CounterFact | | | ZsRE | | | FEVER | | |
|---|---|---|---|---|---|---|---|---|---|---|
| | | Eff.↑ | Gen.↑ | Spe.↑ | Eff.↑ | Gen.↑ | Spe.↑ | Eff.↑ | Gen.↑ | Spe.↑ |
| Llama-3 | **RLEdit** | 91.75 | 62.40 | 52.38 | 88.65 | 83.91 | 47.61 | 94.46 | 91.56 | 69.01 |
| | w/o RL training framework | 50.30↓41.45 | 49.55↓12.85 | 49.63↓2.75 | 0.62↓88.03 | 0.53↓83.38 | 2.36↓45.25 | 21.28↓73.18 | 21.15↓70.41 | 13.63↓55.38 |
| | w/o memory backtracking | 89.84↓1.91 | 60.42↓1.98 | 50.23↓2.15 | 88.62↓0.03 | 81.23↓2.68 | 44.43↓3.18 | 93.02↓1.44 | 87.29↓4.27 | 68.87↓0.14 |
| | w/o regularization | 92.12↑0.37 | 60.99↓1.41 | 52.45↑0.07 | 88.63↓0.02 | 83.12↓0.79 | 48.43↑0.82 | 94.01↓0.45 | 91.58↑0.02 | 68.68↓0.33 |
| Gemma-2 | **RLEdit** | 90.11 | 61.34 | 48.57 | 89.22 | 79.85 | 35.57 | 95.13 | 91.70 | 71.83 |
| | w/o RL training framework | 18.13↓71.98 | 20.73↓40.61 | 80.49↑31.92 | 12.98↓76.24 | 10.49↓69.36 | 9.12↓26.45 | 0.00↓95.13 | 0.00↓91.70 | 0.00↓71.83 |
| | w/o memory backtracking | 89.37↓0.74 | 60.93↓0.41 | 48.68↑0.11 | 87.13↓2.09 | 78.29↓1.56 | 35.27↓0.30 | 95.10↓0.03 | 90.98↓0.72 | 69.28↓2.55 |
| | w/o regularization | 89.87↓0.24 | 62.03↑0.69 | 48.74↑0.17 | 89.21↓0.01 | 77.25↓2.60 | 33.60↓1.97 | 94.95↓0.18 | 92.33↑0.63 | 73.98↑2.15 |
| Mistral | **RLEdit** | 84.24 | 63.93 | 60.79 | 84.60 | 78.00 | 50.18 | 97.78 | 96.34 | 83.71 |
| | w/o RL training framework | 50.95↓33.29 | 49.93↓14.00 | 51.11↓9.68 | 6.66↓77.94 | 6.58↓71.42 | 2.10↓48.08 | 46.38↓51.40 | 45.26↓51.08 | 30.16↓53.55 |
| | w/o memory backtracking | 82.14↓2.10 | 60.51↓3.42 | 58.99↓1.80 | 84.61↑0.01 | 76.89↓1.11 | 48.47↓1.71 | 97.10↓0.68 | 94.29↓2.05 | 81.99↓1.72 |
| | w/o regularization | 84.31↑0.07 | 63.89↓0.04 | 59.88↓0.91 | 85.17↑0.57 | 77.73↓0.27 | 48.21↓1.97 | 97.79↑0.01 | 96.28↓0.06 | 82.92↓0.79 |

As observed, the RL training framework in RLEdit is crucial for adapting hypernetworks to lifelong editing. After ablating the RL training framework, RLEdit's performance metrics significantly decrease, essentially losing its lifelong editing capability. Memory backtracking effectively enhances long-sequence editing performance, as evidenced by the decline in Efficacy and Generalization metrics when it is ablated. Regularization primarily serves to maintain LLM's general capabilities after sequential editing, as ablating regularization results in a decrease in the post-edited LLM's general capabilities.

## B.2. More Results of Editing with Varying Numbers of Knowledge

To evaluate the generalization capabilities of RLEdit across various editing tasks and sequence lengths, we conducted additional experiments. Specifically, we examined the editing performance on tasks of dimensions 20×100, 50×100, 100×100, and 150×100 using Llama-3-8B on three datasets[2], as shown in Table 5.

Comparisons with baseline methods demonstrate that RLEdit maintains relatively stable editing performance across knowledge sequences of any length, while most baselines lose their effectiveness beyond 5,000 samples.

## B.3. General Capability Test on CounterFact

To comprehensively analyze RLEdit's effectiveness in maintaining LLM's general capabilities, to complement the downstream task evaluations on ZsRE presented in Section 4.5, we conducted GLUE task testing on CounterFact for all methods. All experiments were performed on LLama-3-8B, as illustrated in Figure 7.

The results demonstrate that on CounterFact, LLMs edited with RLEdit for 3,000 knowledge samples maintained general capabilities comparable to the pre-trained model, whereas most other methods showed significant degradation in performance.

## B.4. Impact of Different Edit Frequencies and Batch Sizes on RLEdit

In lifelong editing, the frequency of edits and batch size are crucial parameters that significantly impact editing success rates. In this section, we investigate how these parameters affect RLEdit's performance. Using the ZsRE dataset, we edited 5,000 knowledge samples on Llama-3-8B with seven different configurations: 10×500, 25×200, 50×100, 100×50, 200×25,

---

[2]Since RLEdit and other hypernetwork-based methods require at least half of the dataset as training data, and given the limited size of the CounterFact dataset, we could only test knowledge editing up to 10,000 samples in the table.

Table 5: Lifelong editing results under different numbers of edited knowledge samples. ♠ denotes locate-then-edit methods while ♥ denotes hypernetwork-based methods. The best results are highlighted in bold, while the second-best results are underlined.

| TASK | DATASET / METHOD | COUNTERFACT EFF.↑ | GEN.↑ | SPE.↑ | ZSRE EFF.↑ | GEN.↑ | SPE.↑ | FEVER EFF.↑ | GEN.↑ | SPE.↑ |
|---|---|---|---|---|---|---|---|---|---|---|
| 20 × 100 = 2000 | FT | 83.33±0.37 | 67.79±0.40 | 46.63±0.37 | 30.54±0.27 | 30.29±0.27 | 15.49±0.18 | 7.35±0.18 | 6.00±0.16 | 24.10±0.15 |
| | MEND♥ | 49.20±0.42 | 50.10±0.33 | 50.05±0.23 | 0.00±0.00 | 0.00±0.00 | 0.00±0.00 | 37.75±0.21 | 37.78±0.30 | 29.10±0.27 |
| | ROME♠ | 64.45±0.37 | 61.42±0.44 | 49.46±0.40 | 2.00±0.11 | 1.68±0.15 | 0.68±0.07 | – | – | – |
| | MEMIT♠ | 65.05±0.48 | 64.68±0.43 | 52.33±0.39 | 57.72±0.37 | 52.48±0.37 | 25.78±0.22 | – | – | – |
| | PRUNE♠ | 68.25±0.28 | 64.75±0.01 | 49.82±0.24 | 24.77±0.37 | 23.87±0.03 | 20.69±0.43 | – | – | – |
| | RECT♠ | 64.00±0.48 | 61.20±0.43 | 60.88±0.37 | 86.02±0.24 | 81.81±0.27 | 32.04±0.23 | – | – | – |
| | AlphaEdit♠ | **98.90±0.10** | **94.22±0.19** | **67.88±0.29** | **94.47±0.13** | **91.13±0.19** | 32.55±0.22 | – | – | – |
| | **RLEdit♥** | 91.75±0.21 | 62.40±0.18 | 52.38±0.19 | 88.65±0.19 | 83.91±0.24 | **47.61±0.23** | 94.46±0.18 | 91.56±0.29 | 69.01±0.29 |
| 50 × 100 = 5000 | FT | 82.10±0.19 | 66.08±0.29 | 40.35±0.36 | 21.05±0.24 | 20.80±0.24 | 9.69±0.14 | 8.40±18.72 | 6.01±16.38 | 23.69±0.15 |
| | MEND♥ | 49.71±0.32 | 49.63±0.33 | 50.23±0.49 | 0.00±0.00 | 0.00±0.00 | 0.00±0.00 | 33.49±0.31 | 28.26±0.22 | 26.77±0.35 |
| | ROME♠ | 48.62±0.50 | 49.78±0.48 | 51.65±0.46 | 1.25±0.11 | 1.30±0.11 | 1.65±0.06 | – | – | – |
| | MEMIT♠ | 64.21±0.23 | 60.07±0.43 | 46.64±0.37 | 0.07±0.01 | 0.07±0.01 | 31.67±0.22 | – | – | – |
| | PRUNE♠ | 56.07±0.34 | 54.58±0.39 | 50.32±0.28 | 0.09±0.01 | 0.08±0.01 | 1.35±0.04 | – | – | – |
| | RECT♠ | 52.64±0.50 | 50.02±0.46 | 53.59±0.42 | 0.07±0.01 | 0.07±0.01 | 1.34±0.03 | – | – | – |
| | AlphaEdit♠ | **94.66±0.15** | **92.35±0.22** | **62.00±0.31** | **93.76±0.15** | **88.65±0.21** | 31.71±0.22 | – | – | – |
| | **RLEdit♥** | 85.54±0.43 | 60.96±0.27 | 48.60±0.29 | 85.46±0.39 | 80.68±0.44 | 43.35±0.21 | 94.37±0.18 | 91.84±0.22 | 67.58±0.27 |
| 100 × 100 = 10000 | FT | 79.99±0.20 | 62.02±0.29 | 38.86±0.36 | 14.80±0.22 | 14.50±0.22 | 5.22±0.10 | 23.02±0.28 | 8.38±0.19 | 22.16±0.16 |
| | MEND♥ | 47.24±0.32 | 47.19±0.23 | **53.10±0.35** | 0.00±0.00 | 0.00±0.00 | 0.00±0.00 | 27.24±0.27 | 28.93±0.32 | 27.56±0.34 |
| | ROME♠ | 46.43±0.39 | 40.99±0.31 | 46.68±0.23 | 1.87±0.12 | 1.84±0.12 | 1.65±0.06 | – | – | – |
| | MEMIT♠ | 47.53±0.50 | 43.59±0.49 | 51.30±0.48 | 0.00±0.00 | 0.00±0.00 | 0.00±0.00 | – | – | – |
| | PRUNE♠ | 48.29±0.23 | 48.01±0.37 | 50.91±0.43 | 0.09±0.01 | 0.06±0.02 | 1.26±0.09 | – | – | – |
| | RECT♠ | 49.95±0.29 | 50.11±0.32 | 46.27±0.25 | 0.08±0.01 | 0.08±0.01 | 1.36±0.04 | – | – | – |
| | AlphaEdit♠ | 80.17±0.21 | **75.34±0.45** | 52.78±0.39 | **89.60±0.17** | **86.75±0.23** | 30.96±0.22 | – | – | – |
| | **RLEdit♥** | **80.69±0.23** | 62.31±0.27 | 45.16±0.31 | 86.45±0.32 | 83.07±0.19 | 41.96±0.34 | 95.06±0.21 | 92.26±0.18 | 69.19±0.30 |
| 150 × 100 = 15000 | FT | **89.79±0.43** | **70.05±0.32** | 36.64±0.22 | 13.95±0.20 | 13.49±0.20 | 3.93±0.08 | 44.84±0.43 | 32.19±0.42 | 33.54±0.41 |
| | MEND♥ | – | – | – | 0.00±0.00 | 0.00±0.00 | 0.00±0.00 | 21.70±0.43 | 20.30±0.24 | 13.86±0.25 |
| | ROME♠ | 46.51±0.50 | 47.10±0.47 | **53.36±0.45** | 1.47±0.11 | 1.43±0.11 | 0.72±0.06 | – | – | – |
| | MEMIT♠ | 52.07±0.45 | 50.23±0.48 | 48.89±0.47 | 0.00±0.00 | 0.00±0.00 | 0.00±0.00 | – | – | – |
| | PRUNE♠ | 52.19±0.24 | 52.87±0.39 | 44.43±0.27 | 0.00±0.00 | 0.00±0.00 | 0.00±0.00 | – | – | – |
| | RECT♠ | 54.49±0.30 | 53.24±0.31 | 46.22±0.27 | 0.02±0.01 | 0.03±0.01 | 1.02±0.21 | – | – | – |
| | AlphaEdit♠ | 79.35±0.35 | 69.28±0.17 | 42.01±0.35 | 84.43±0.25 | 78.28±0.30 | 28.48±0.21 | – | – | – |
| | **RLEdit♥** | – | – | – | 87.81±0.43 | 85.13±0.22 | 40.98±0.37 | 95.11±0.26 | 91.85±0.24 | 68.59±0.29 |
| 200 × 100 = 20000 | FT | **86.23±0.49** | **69.77±0.31** | 38.94±0.33 | 13.21±0.30 | 11.11±0.27 | 4.24±0.15 | 55.89±0.37 | 50.07±0.52 | 49.79±0.30 |
| | MEND♥ | - | - | – | 0.00±0.00 | 0.00±0.00 | 0.00±0.00 | 11.08±0.27 | 8.01±0.11 | 15.38±0.53 |
| | ROME♠ | 44.43±0.39 | 46.21±0.27 | **50.77±0.33** | 1.56±0.21 | 1.34±0.19 | 1.04±0.08 | – | – | – |
| | MEMIT♠ | 49.98±0.29 | 46.55±0.30 | 45.23±0.33 | 0.00±0.00 | 0.00±0.00 | 0.00±0.00 | – | – | – |
| | PRUNE♠ | 50.21±0.22 | 47.43±0.32 | 47.11±0.22 | 0.00±0.00 | 0.00±0.00 | 0.00±0.00 | – | – | – |
| | RECT♠ | 43.32±0.29 | 42.21±0.37 | 42.01±0.20 | 0.00±0.00 | 0.00±0.00 | 0.00±0.00 | – | – | – |
| | AlphaEdit♠ | 74.22±0.57 | 63.94±0.46 | 39.12±0.45 | 77.74±0.42 | 70.01±0.34 | 27.96±0.28 | – | – | – |
| | **RLEdit♥** | – | – | – | 91.38±0.43 | 89.93±0.29 | 41.98±0.30 | 94.03±0.37 | 90.67±0.42 | 68.71±0.41 |

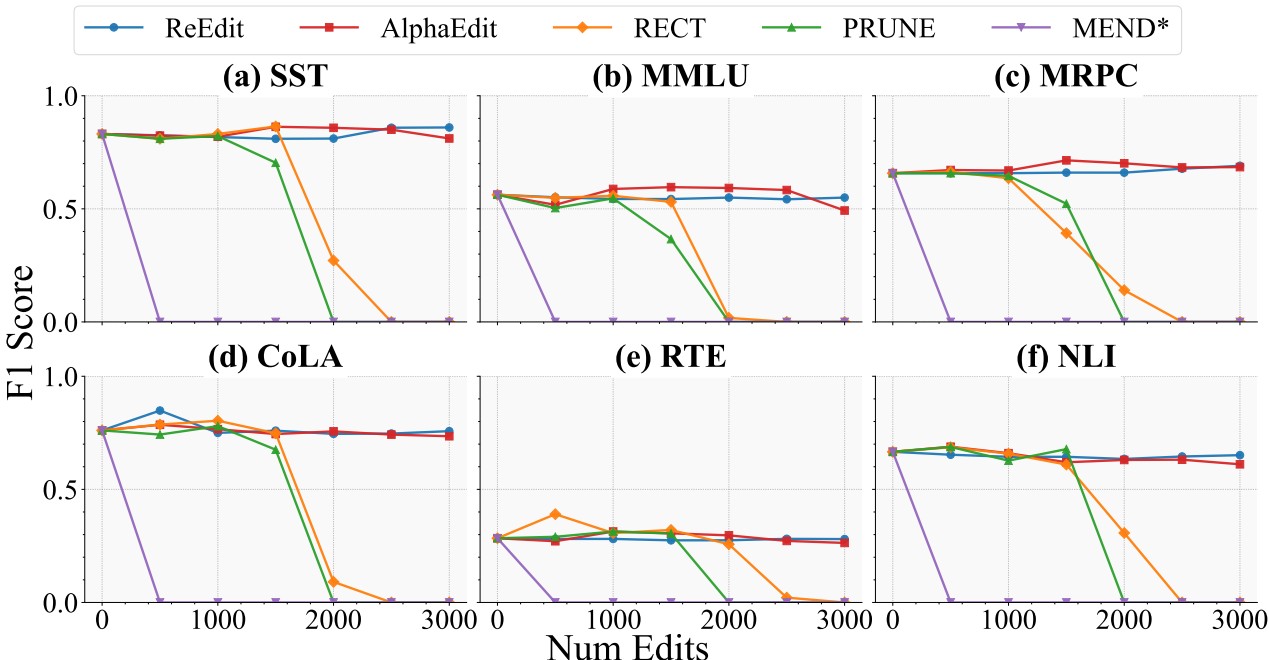

Figure 7: General Capability Test on CounterFact.

500×10, and 1000×5. Figure 8 shows how various metrics vary across these configurations.

From Figure 8, we observe that RLEdit demonstrates excellent editing performance across all configurations. However, we find that under extreme conditions with very large batch sizes, the hypernetwork training in RLEdit consumes substantial GPU memory, leading to high resource utilization. Similarly, under extreme conditions with very high edit frequencies, the hypernetwork training becomes unstable and often fails to converge. These observations suggest that appropriate edit frequencies and batch sizes should be selected to achieve more effective and stable editing results.

## C. Algorithms

In Sections 3.2 and Section 3.3, we introduced the hypernetwork training and editing procedures of RLEdit. The corresponding pseudocode for RLEdit's editing algorithms is presented in Algorithm 2.

---
**Algorithm 2** RLEdit Editing

---
**Input:** Knowledge sequence $(x_i, y_i)_{i=1}^n \in \mathcal{D}$, pre-trained LLM $f_{\mathcal{W}_0}$, post-trained hypernetwork $\mathcal{H}$ with parameter $\theta'$, hyper-parameters $k$, $\gamma$, $\lambda_{loc}$ and $\eta$
**Output:** Post-edited LLM with parameters $\mathcal{W}_n$
**for** $t = 1$ **to** $n$ **do**
    $\mathcal{L}_t \leftarrow -\log p_{\mathcal{W}_{t-1}}(y_t \,|x_t)$
    Back-propagate $\mathcal{L}_t$ and cache $\nabla_{\mathcal{W}_{t-1}}$
    $\tilde{\nabla}_{\mathcal{W}_t} \leftarrow \mathcal{H}(\nabla_{\mathcal{W}_{t-1}})$
    $\mathcal{W}_t \leftarrow \mathcal{W}_{t-1} + \tilde{\nabla}_{\mathcal{W}_t}$
**end for**
**return** $\mathcal{W}_n$

---

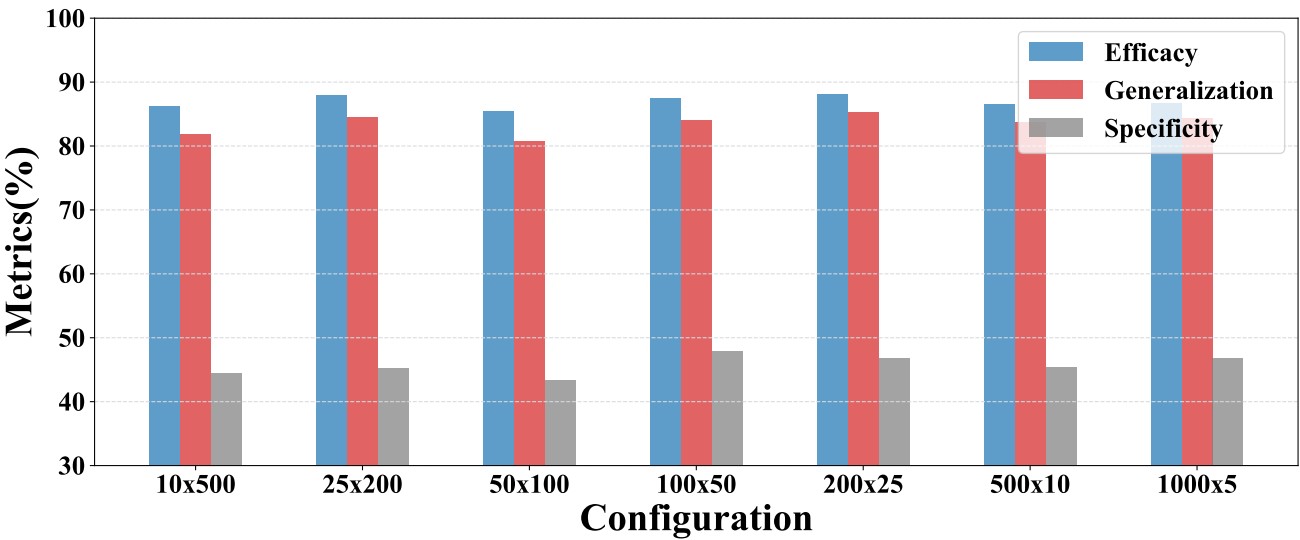

Figure 8: RLEdit's final metric scores when editing 5,000 knowledge instances under 7 different configurations.

## D. Detailed Proof

In Section 3.1, we briefly introduced how hypernetwork-based lifelong editing can be modeled as an MDP. Here we provide detailed proof that hypernetwork-based lifelong editing is indeed an MDP.

In Section 3.1.1, we stated that hypernetwork-based lifelong editing is a Markov process (MP). An MP consists of several elements: (1) a well-defined state space, (2) stationary state transitions, and (3) satisfaction of probability axioms. For (1), we model the state as $(\mathcal{W}, (x, y))$ in hypernetwork-based lifelong editing, or specifically, the fine-tuning gradient $\nabla_\mathcal{W}$. This state is defined in a continuous parameter space that can take tensors of fixed dimensions. Each state has a concrete mathematical representation. For (2), we modify LLM parameters through hypernetwork-generated parameter updates, corresponding to state transitions in MDP; when we add Gaussian noise from a fixed distribution to the hypernetwork output, our transition function's probability distribution becomes stationary. For (3), Gaussian noise satisfies probability axioms (probabilities are non-negative and sum to 1), thus state transitions also satisfy these axioms. In conclusion, hypernetwork-based lifelong editing perfectly aligns with MP modeling and its characteristics.

We now prove that this MP is an MDP. Compared to MP, MDP introduces concepts of action space, state transition function, and reward function. Regarding action space, hypernetwork-based lifelong editing's actions are continuous parameter change matrices with dimensions matching the parameter matrices, which are fixed; since actions are controlled through hypernetwork parameters, they satisfy controllability; parameter change matrices in finite-dimensional real space possess complete metric structure, thus are well-defined. For the state transition function, we have already proven it is well-defined and satisfies probability axioms. The non-deterministic component in the state transition function is Gaussian noise, whose distribution remains unchanged over time, proving state transition stationarity. Regarding the reward function, Section 3.1 explicitly detailed its computation method. Since rewards are derived from losses that have bounds, the reward function is bounded; furthermore, the computation method remains constant over time, demonstrating stability. Therefore, hypernetwork-based lifelong editing process constitutes a standard MDP.

## E. More Discussion

RLEdit training achieves impressive results with relatively few trajectory samples. This suggests that the lifelong editing task may be simpler than classic RL tasks, while also demonstrating RLEdit's strong generalization capability. This insight inspired us to explore simpler few-shot knowledge editing paradigms.

In our experiments, we found that different choices of LLM editing layers significantly impact editing effectiveness. While locate-then-edit methods (Meng et al., 2023) typically choose to edit the first few layers of LLM due to their optimal editing performance, we discovered that RLEdit performs best when editing the middle and final layers of LLM. We believe this insight motivates further research on lifelong editing from the perspective of sequential knowledge localization. Additionally, we will investigate the use of advanced approaches of LLM interpretability (He et al., 2024; Zhou et al., 2024) to identify and

extract crucial layers within the model, enabling targeted editing and refinement for improved performance and adaptability.

Furthermore, in the future, we aim to apply the reinforcement learning paradigm to a broader range of model editing methods, such as locate-and-edit (Jiang et al., 2025) and memory-based (Wang et al., 2024) approaches. Additionally, to address the challenges of updating multi-hop and related knowledge, we plan to enhance RLEdit by integrating external knowledge bases or knowledge graphs (Chen et al., 2023; 2024b; Zhang et al., 2025b;a), which is crucial for further advancements. We will continue to investigate the hallucination and security issues (Chen et al., 2024a; Yang et al., 2024a;b; Liu et al., 2024; Huang et al., 2025) brought by model editing, striving to enhance the reliability and safety of this technology to achieve more responsible AI development.

# F. Dataset Visualization through Examples

To help readers better understand the lifelong editing task and our implementation approach, we provide 3 examples from the ZsRE, FEVER, and CounterFact datasets, as illustrated in Figures 9, 10, and 11.

```
{
        "subject": "Lahti Town Hall",
        "src": "Which person is the architect of Lahti Town Hall?",
        "rephrase": "Who was the architect of Lahti Town Hall?",
        "alt": "Willem Marinus Dudok",
        "loc": "nq question: who plays alec ramsay in the black stallion",
        "loc_ans": "Kelly Reno",
        "ans": "Eliel Saarinen"
},
{
        "subject": "Phantom Stranger",
        "src": "In which fictional universe is Phantom Stranger a character?",
        "rephrase": "In what fictitious universe is Phantom Stranger part?",
        "alt": "Forgotten Realms",
        "loc": "nq question: who was the successful commanding general of the northern forces in the civil war",
        "loc_ans": "George B. McClellan",
        "ans": "DC Universe"
},
{
        "subject": "1963 BAC One-Eleven test crash",
        "src": "What is the full date that 1963 BAC One-Eleven test crash crashed on?",
        "rephrase": "What's the date on which 1963 BAC One-Eleven test crash happened?",
        "alt": "11 March 1963",
        "loc": "nq question: what is the current rate of interest on ppf",
        "loc_ans": "7.6%",
        "ans": "22 October 1963"
}
```

Figure 9: A sample of ZsRE dataset.

```
{
    "prompt": "The Faroe Islands were part of the Hereditary Kingdom of Norway between 1035 and 1814.",
    "equiv_prompt": [
        "The Faroe Islands were part of the Kingdom of Norway between 1035 and 1814.",
        "The Faroes were part of the Kingdom of Norway between 1035 and 1814.",
        "The Faroe Islands were part of the Kingdom of Norway from 1035 to 1814.",
        "The Faroe Islands were part of Norway between 1035 and 1814.",
        "The Faroe Islands were part of the Hereditary Kingdom of Norway from 1035 to 1814."
    ],
    "ans": "SUPPORTS",
    "alt": "SUPPORTS",
    "unrel_prompt": "Prague's historic center was first included in the UNESCO list of World Heritage Sites in the nineties of the twentieth century.",
    "unrel_ans": "SUPPORTS"
}
```

Figure 10: A sample of FEVER dataset.

```
{
  "case_id": 7285,
  "pararel_idx": 18788,
  "requested_rewrite": {
    "prompt": "The language of {} is",
    "relation_id": "P407",
    "target_new": {
      "str": "Norwegian",
      "id": "Q9043"
    },
    "target_true": {
    "str": "Dutch",
    "id": "Q7411"
    },
    "subject": "Goede Tijden, Slechte Tijden"
  },
  "paraphrase_prompts": [
    "The story is an adaptation of Swan Lake. Goede Tijden, Slechte Tijden was written in",
    "Winnie YU Tsang (\u4fde\u7424, b. Goede Tijden, Slechte Tijden is written in"
  ],
  "neighborhood_prompts": [
    "Counter-Strike: Global Offensive was written in",
     "Marie Claire is written in",
    "The language of Magdalena was",
    "Adobe Flash Player was written in",
    "The language of Marie Claire is",
    "The language of Eva is",
    "Betsy was written in",
    "The language of Ben is",
    "The language of Treaty of Accession 2003 was",
    "The language of Treaty of Accession 2003 is"
  ],
  "attribute_prompts": [
    "The language of Kristin Lavransdatter is",
    "Kristin Lavransdatter is written in",
    "The language of 20 \u00e5r med oss \u2013 Vem \u00e9 d\u00e9 du vill ha is",
    "Mikkel is written in",
    "For all tid was written in",
    "The language of Klassekampen is",
    "Ove is written in",
    "The language of Ghosts was",
    "The language of Jakobsen is",
    "The language of Morgenbladet was"
  ],
  "generation_prompts": [
    "Goede Tijden, Slechte Tijden's author is from",
    "Goede Tijden, Slechte Tijden's author is from",
    "The language Goede Tijden, Slechte Tijden was written in is called",
    "Goede Tijden, Slechte Tijden's author is from",
    "Goede Tijden, Slechte Tijden's author is from",
    "The language Goede Tijden, Slechte Tijden was written in is called",
    "The language Goede Tijden, Slechte Tijden was written in is called",
    "The language Goede Tijden, Slechte Tijden was written in is called",
    "Goede Tijden, Slechte Tijden's author is from",
    "The language Goede Tijden, Slechte Tijden was written in is called"
  ]
}
```

Figure 11: A sample of CounterFact dataset.

