# OpenReview forum: "Reinforced Lifelong Editing for Language Models"
_ICML.cc/2025/Conference — ICML 2025 poster_

### Official Review · Reviewer_kk6q · 2025-03-02

**Overall Recommendation:** 4

**Summary:**

This paper introduces RLEdit, a method addressing long-term continuous LLM knowledge editing, known as lifelong editing. The authors apply reinforcement learning modeling to meta-learning knowledge editing approaches, proposing a hypernetwork training method specifically designed for lifelong editing tasks. They focus on addressing knowledge forgetting and conflicts in lifelong editing by  designing a reward function with memory backtracking to optimize hypernetwork performance on long sequence editing. Experiments demonstrate the method's effectiveness across various configurations while maintaining high efficiency.

**Claims And Evidence:**

Yes

**Essential References Not Discussed:**

None

**Experimental Designs Or Analyses:**

Extensive experiments validate RLEdit's high performance and efficiency, with the authors providing code and detailed hyperparameter configurations to ensure reproducibility.

**Methods And Evaluation Criteria:**

Yes.This paper presents the first attempt to model hypernetwork-based lifelong editing using reinforcement learning, demonstrating remarkable novelty.

**Other Comments Or Suggestions:**

None

**Other Strengths And Weaknesses:**

Equation (9) requires more clarification. A detailed explanation would help readers better understand this notation.

Minor: Some typos. (*e.g.*, Line 97, right column: "updated" should be "updates"; Line 159, left column: "he" should be "The".)

**Questions For Authors:**

Figure 5 only examines the LLM's general capacity after 3,000 edits across different methods. Given that you've extended lifelong editing to over 10,000 edits, I believe the general capacity evaluation should be expanded to cover more editing steps. Have you conducted related experiments?

I'm curious about how the choice of LLM layers affects model editing performance. Have you explored varying the selection and number of edited LLM layers and investigated their impact on the results?

**Relation To Broader Scientific Literature:**

This paper presents the first attempt to model hypernetwork-based lifelong editing using reinforcement learning, demonstrating remarkable novelty，Previous articles usually start from the initial state.

**Theoretical Claims:**

Yes,all of them

---

> ### Author Rebuttal · Authors · 2025-03-28
>
> Dear Reviewer kk6q:
>
> **Thank you for your positive feedback on our work!**
>
> We are happy to discuss the details of our work and hope to address all your questions.
>
> ******
>
> ## For *Other Strengths And Weaknesses*:
>
> > Equation (9) requires more clarification. A detailed explanation would help readers better understand this notation.
>
> Thank you for your correction! The terms $\mathcal{L} _ {e _ i, \mathcal{W} _ {t-1}}$ and $\mathcal{L} _ {loc _ {i}, \mathcal{W} _ {t-1}}$ in the equation represent the loss functions corresponding to the editing knowledge at time step $i$, computed on the model parameters $\mathcal{W} _ {t-1}$ at time step $t-1$.
>
> We agree that Equation (9) indeed needs clearer explanation, and we will add detailed clarification in the revised version to enhance readability.
>
> > Minor: Some typos. (*e.g.*, Line 97, right column: "updated" should be "updates"; Line 159, left column: "he" should be "The".)
>
> Thank you for identifying these typos! We will correct these issues in the revised version, including changing "updated" to "updates" in the right column of line 97 and "he" to "The" in the left column of line 159.
>
> ## For *Questions For Authors*:
>
> > Figure 5 only examines the LLM's general capacity after 3,000 edits  across different methods. Given that you've extended lifelong editing to over 10,000 edits, I believe the general capacity evaluation should be  expanded to cover more editing steps. Have you conducted related experiments?
>
> We fully agree with your suggestion. To verify that RLEdit can maintain LLM's general capabilities after large-scale continuous editing, it is necessary to evaluate the model's performance after completing 10,000 edits.
>
> Following your suggestion, we have supplemented our experiments to assess the impact of large-scale model editing on the model's general capabilities. Consistent with Section 4.5 of our paper, we used a 200$\times$100 configuration (*i.e.*, 200 batches with 100 edits per batch) to edit Llama-3-8B-Instruct model on the ZsRE dataset. We conducted GLUE tests after every 2,000 edits, recording F1 Scores for SST, MMLU, MRPC, CoLA, RTE, and NLI metrics. The experimental results are as follows:
>
> |       | SST    | MMLU   | MRPC   | CoLA   | RTE    | NLI    |
> | :---- | ------ | ------ | ------ | ------ | ------ | ------ |
> | 0     | 0.8311 | 0.5624 | 0.6576 | 0.7607 | 0.2839 | 0.6660 |
> | 2000  | 0.8791 | 0.4944 | 0.6862 | 0.7703 | 0.2809 | 0.5924 |
> | 4000  | 0.8797 | 0.5010 | 0.7093 | 0.7703 | 0.2870 | 0.5685 |
> | 6000  | 0.8797 | 0.4853 | 0.7190 | 0.7594 | 0.2809 | 0.5685 |
> | 8000  | 0.8779 | 0.5019 | 0.7097 | 0.7498 | 0.2870 | 0.5328 |
> | 10000 | 0.8654 | 0.4707 | 0.7196 | 0.7607 | 0.2870 | 0.5907 |
> | 12000 | 0.8654 | 0.4759 | 0.7278 | 0.7423 | 0.2930 | 0.6044 |
> | 14000 | 0.8381 | 0.4599 | 0.7300 | 0.7389 | 0.3056 | 0.5907 |
> | 16000 | 0.8443 | 0.4805 | 0.7298 | 0.7510 | 0.2967 | 0.5907 |
> | 18000 | 0.8456 | 0.4438 | 0.7383 | 0.7617 | 0.3030 | 0.5907 |
> | 20000 | 0.8395 | 0.4653 | 0.7391 | 0.7437 | 0.3030 | 0.5907 |
>
> From the experimental data, we can observe that as the number of knowledge samples continues to increase, our RLEdit method consistently maintains stable general capabilities of the LLM. Notably, even after 20,000 knowledge edits, the model maintains general capability metrics comparable to its initial state. This strongly demonstrates RLEdit's exceptional stability in large-scale continuous  knowledge editing scenarios.
>
> > I'm curious about how the choice of LLM layers affects model editing  performance. Have you explored varying the selection and number of  edited LLM layers and investigated their impact on the results?
>
> In our experiments, we found that the choice of editing layers significantly impacts the editing effectiveness. Specifically, when selecting early layers for RLEdit, the final metrics might decrease by 5%-10% compared to selecting middle or later layers; certain layer choices might result in very high Efficacy but very low Generalization and Specificity.
>
> Therefore, the selection of LLM layers is a hyperparameter that requires careful tuning, which also demonstrates the complexity of LLM knowledge storage mechanisms.
>
> Currently, several studies are focusing on investigating the impact of layer selection on editing effectiveness. We will include references to these works and add discussion of this issue in our subsequent revision.
>
> ******
>
> Thank you again for your positive feedback and valuable suggestions on our work! If you have any other questions, we would be happy to continue the discussion.
>
> Best regards
>
> Authors

---

> > ### Comment · Reviewer_kk6q · 2025-04-01
> >
> > The author's rebuttal effectively reduce my concerns, and I raise my score

---

### Official Review · Reviewer_TQMf · 2025-03-10

**Overall Recommendation:** 4

**Summary:**

The paper proposes a model editing method, named RLEdit, for lifelong editing that extends meta-learning methods to sequential editing tasks. RLEdit's hypernetwork is trained using reinforcement learning algorithms, effectively maintaining Effectiveness, Generalization, and Locality at the knowledge sequence level. Experiments demonstrate that RLEdit shows advantages in both editing success rate and efficiency.

**Claims And Evidence:**

The authors' claims about RLEdit are substantiated through rigorous mathematical derivations and comprehensive experiments. The training of RLEdit's hypernetwork, based on reinforcement learning modeling and utilizing policy gradient concepts to optimize carefully designed reward functions, demonstrates both convincing theoretical foundation and practical implementability. The evaluation across various models and datasets further validates the credibility of their conclusions.

**Essential References Not Discussed:**

The paper provides fairly comprehensive citations of landmark papers in the lifelong model editing field. However, I believe the authors could include more references to seminal works in reinforcement learning to strengthen the theoretical validation. Additionally, some recent lifelong editing methods (*e.g.*, GLAME, O-Edit) could also be discussed.

**Experimental Designs Or Analyses:**

The experimental design in the paper is comprehensive, covering 3 commonly used open-source LLMs (Llama3, Gemma2, Mistral) and 3 standard datasets in the model editing field (ZsRE, FEVER, CounterFact), demonstrating RLEdit's generalization ability and robustness across different scenarios. The authors also compare RLEdit with numerous baseline methods, proving RLEdit's superior performance and efficiency.

However, one concern stands out: Figure 5 does not show the LLM's general capability metrics when editing more than 3k knowledge items. The authors should supplement related experiments to demonstrate RLEdit's robustness in editing 10k knowledge items.

**Methods And Evaluation Criteria:**

Yes. The authors designed a targeted solution for lifelong editing tasks, achieving long knowledge sequence lifelong editing through the application of the RLEdit hypernetwork for continuous editing. This establishes a new editing paradigm while maintaining both effectiveness and efficiency. The evaluation methods used in the paper are standard in the model editing field, such as examining Effectiveness,  Generalization, and Specificity on CounterFact, Zsre, and FEVER datasets, making their results convincing.

**Other Comments Or Suggestions:**

There are some minor typos in the paper. For example, I believe the $\nabla_W$ in line 117 should be $\tilde{\nabla}_\mathcal{W}$, and "he" in line 159 should be "The".

**Other Strengths And Weaknesses:**

# Strengths

1. The paper presents an innovative approach to long-sequence lifelong editing, addressing a practically significant problem. I agrees with authors' perspective that continuous knowledge editing requires modeling at the entire knowledge sequence level, which is an elegant and holistic idea.
2. Based on this, the authors proposes an effective and efficient editing method. To my knowledge, it is the first to extend meta-learning editing to continuous editing tasks at 10k scale, which demonstrates pioneering work in hypernetwork-based large-scale knowledge editing.
3. RLEdit shows strong transferability on MEND and MALMEN, revealing the potential of meta-learning methods in lifelong editing. This observation provides valuable insights for future research.
4. The comprehensive experiments demonstrates RLEdit's versatility across multiple scenarios.

# Weaknesses

1. Comparing RLEdit with parameter-preserving editing methods (*e.g.*, GRACE, SERAC) would be more convincing, as they also target lifelong editing. This is only a suggestion and will not affect my rate.
2. As mentioned in **Experimental Designs Or Analyses**, tests of LLM's general capabilities after editing more than 3k knowledge items should be conducted to demonstrate RLEdit's effectiveness at the 10k scale of knowledge editing.

**Questions For Authors:**

1. In Table 2, many editing methods fail under specific LLMs, with metrics dropping to as low as 0.00%. Have you investigated the reasons for this?
2. Why don't MEND\* and MALMEN\* show significant improvements over MEND  and MALMEN? I feel that retraining the hypernetwork on post-edited LLMs after each edit might be effective, although highly inefficient.
3. In Figure 8, I'm surprised by RLEdit's consistently strong performance across different configurations. Have you explored how other baseline methods (*e.g.*, MEMIT, AlphaEdit) perform under different batch_size and batch_number ratios? How do you explain RLEdit's stability?

**Relation To Broader Scientific Literature:**

RLEdit proposed in this paper marks a significant advancement in extending meta-learning model editing methods to lifelong editing of long knowledge sequences. RLEdit optimizes existing meta-learning editing methods and can be generalized to MEND and MALMEN, endowing them with lifelong editing capabilities, thus providing novel and valuable insights for the field of model editing.

**Theoretical Claims:**

In this paper, the authors propose a novel idea of using reinforcement learning to model the training process of editing hypernetworks. The authors clearly demonstrate that meta-learning editing methods can be modeled as an MDP,  making them solvable through reinforcement learning. They then design reward functions targeting the key challenges of lifelong editing, with solid and well-founded theoretical derivations. I have no concerns regarding the theoretical aspects of this paper.

---

> ### Author Rebuttal · Authors · 2025-03-28
>
> Dear reviewer TQMf:
>
> **Thank you very much for your recognition and support of our work!**
>
> We are very willing to engage in in-depth discussions with you regarding the research details.
>
> ******
>
> ## For *Experimental Designs Or Analyses*:
>
> Thank you for your valuable suggestions!
>
> Following your suggestion, we have conducted additional experiments. Due to the word limit constraints of the rebuttal, detailed experimental results can be found in **our response to Reviewer kk6q**.
>
> The results shows that even after completing 20k knowledge edits, the model maintained stable GLUE scores. We will include these experimental data in the revised paper to ensure evaluation completeness.
>
> ## For *Essential References Not Discussed*:
>
> We strongly agree with your suggestion.
>
> In the revised version, we will add citations to key literature on RL theoretical foundations and supplement discussions of recent research developments in lifelong editing methods.
>
> ## For *Weakness 1*:
>
> Thank you very much for your suggestions!
>
> We understand the importance of comparing RLEdit with parameter-preserving editing methods like GRACE and SERAC, although they operate on different mechanisms from RLEdit.
>
> Following your suggestion, we have conducted comparative experiments between RLEdit and GRACE, SERAC on 20$\times$100 tasks. The experiments were performed on Llama-3-8B-Instruct using both ZsRE and CounterFact datasets. The results are shown in the following table:
>
> |        |              |     ZsRE     |              |              | CounterFact  |              |
> | :----: | :----------: | :----------: | :----------: | :----------: | :----------: | :----------: |
> |        |     Eff.     |     Gen.     |     Spe.     |     Eff.     |     Gen.     |     Spe.     |
> | GRACE  |  **93.58**   |     1.03     | <u>31.86</u> |  **96.72**   |    50.14     |  **72.23**   |
> | SERAC  |    67.75     | <u>33.96</u> |    22.17     |    71.21     | <u>61.05</u> | <u>66.90</u> |
> | RLEdit | <u>88.65</u> |  **83.91**   |  **47.61**   | <u>91.75</u> |  **62.40**   |    52.38     |
>
> We observe that RLEdit shows comparable performance to parameter-preserving editing methods, demonstrating the effectiveness of the RLEdit approach.
>
> ## For *Weakness 2*:
>
> Thank you again for this key suggestion!
>
> As stated above, we have added relevant experiments. Detailed results have been presented in **our response to Reviewer kk6q**.
>
> ## For *Other Comments Or Suggestions*:
>
> Thank you very much for your careful review of the paper!
>
> We will carefully revise these errors in the revised manuscript to ensure that the final submitted paper meets high quality standards.
>
> ## For *Question 1*:
>
> Thank you for highlighting this important observation. We have conducted an in-depth analysis of this phenomenon and identified several main reasons:
>
> 1. This high-density, long-sequence editing task significantly amplifies the  cumulative effect of editing errors, potentially triggering catastrophic forgetting or even severely damaging the LLM's basic functionalities.
> 2. Although our baseline comparison methods perform excellently in  single or short-sequence editing scenarios, most were not specifically  optimized for long sequence continuous editing scenarios.
>
> This finding was indeed one of the key motivations that drove us to develop RLEdit, aiming to achieve more precise and robust long-sequence lifelong editing capabilities.
>
> ## For *Question 2*:
>
> Thank you for raising this profound question.
>
> Our analysis suggests this phenomenon is primarily related to inherent limitations in hypernetwork training mechanisms.
>
> Parameter changes and potential errors introduced by the previous batch of edits continue to accumulate and propagate to subsequent edits, and the model instability caused by this error accumulation effect significantly weakens the overall editing performance of these methods.
>
> While complete retraining might theoretically help, as you point out, this indeed poses serious computational efficiency challenges, especially in large-scale lifelong editing scenarios.
>
> ## For *Question 3*:
>
> Thank you for your attention to the results in Figure 8. We indeed systematically explored the performance of various baseline methods under different batch configurations.
>
> Our research found that, with a fixed total number of editing samples, the performance of most baseline methods is highly correlated with specific editing configurations. Particularly, when reducing batch size while increasing batch number, these methods generally show significant performance degradation.
>
> We believe RLEdit's stability stems from its sequence-level perspective. RLEdit hypernetwork training focuses on the entire knowledge sequence rather than individual knowledge samples, enabling the hypernetwork to handle diverse editing configurations.
>
> ******
>
> Thank you again for your detailed and thorough feedback! If you have any other questions, we would be happy to continue the discussion.
>
> Best regards
>
> Authors

---

### Official Review · Reviewer_2E3w · 2025-03-12

**Overall Recommendation:** 3

**Summary:**

The paper proposes RLEdit, a novel Reinforcement Learning based hypernetwork approach to edit Large Language Models for knowledge update after model completes training. It primarily aims to solve the challenge that hypernetwork based editing methods are efficient yet struggle more on large amount of edits. By using the dynamic environmental feedback in RL process, the model can better capture and edit based on the dynamics in the LLM. The work introduces Memory Backtracking Component and Regularization to Reward function in addition to the basic component, knowledge update and preservation objective. The authors show that RLEdit can outperform previous methods both in speed and performance when applied to different LLMs and data, especially when edits increase.

**Claims And Evidence:**

The claims of the paper are clear and well-supported.

**Essential References Not Discussed:**

N/A

**Experimental Designs Or Analyses:**

The experiments look comprehensive.

**Methods And Evaluation Criteria:**

Frankly, I am unfamiliar with the paper's topic on large language model editing. The proposed method is complicated and challenges my knowledge range, so I can only give educated guess.  The method is lifelong editing through training a hypernetwork adapted to knowledge sequences (Meta learning), which is trained using reinforcement learning (RL). I can see that the evaluation follows standard practice and is compared against previous methods.   The paper is well written and clearly organized.

**Other Comments Or Suggestions:**

1. typo: Section 3.1.1 "he training process" should be "the"

**Other Strengths And Weaknesses:**

N/A

**Questions For Authors:**

I wonder compared to RAG, what is the pros and cons using the editing method.

**Relation To Broader Scientific Literature:**

I think Retrieval-Augmented (RAG) is a popular way and drawing the most attention for updating LLM's knowledge.

**Theoretical Claims:**

N/A

---

> ### Author Rebuttal · Authors · 2025-03-28
>
> Dear Reviewer 2E3w:
>
> **Thank you for your positive feedback on our work!**
>
> In brief, we propose a hypernetwork training method for lifelong editing tasks that can adaptively adjust LLM parameters based on knowledge update sequences, achieving efficient and low-interference lifelong editing.
>
> We appreciate your recognition of our paper writing and language organization, and we are happy to further simplify the technical descriptions in the final version to make them more accessible.
>
> *****
>
> ## For *Relation To Broader Scientific Literature*:
>
> > I think Retrieval-Augmented (RAG) is a popular way and drawing the most attention for updating LLM's knowledge.
>
> Thank you for your insights.
>
> Indeed, Retrieval-Augmented Generation (RAG) is a popular technique that effectively enhances LLM's generation capabilities by retrieving relevant information from external vector databases as contextual input. The main value of RAG lies in expanding the model's knowledge scope and improving domain-specific performance, and it can also be used for knowledge editing, thanks to LLM's powerful contextual learning abilities.
>
> However, we believe RAG has several limitations in the field of knowledge editing:
>
> 1. **Effectiveness Challenges**: When correcting outdated or incorrect knowledge stored in LLM parameters, RAG methods inevitably lead to knowledge conflicts. Specifically, contradictions between retrieved new knowledge and model's built-in knowledge significantly reduce editing effectiveness, particularly evident in factual update tasks.
> 2. **Insufficient Persistence**: The core objective of model editing is to permanently modify LLM's internal knowledge representations. Unlike methods that modify model parameters, RAG can be characterized as a prompt tuning technique, where knowledge updates are only effective in specific contexts, making it difficult to achieve persistent model editing.
>
> In Model Editing research, we categorize methods like RAG that add auxiliary modules without changing base model parameters as *parameter-preserving editing methods* (as stated in Appendix B, Line 836). If you're interested in such methods, we recommend referring to the related research cited in our Related Work section.
>
> ## For *Other Comments Or Suggestions*:
>
> > typo: Section 3.1.1 "he training process" should be "the"
>
> Thank you for your thorough review! The typo you identified in Section 3.1.1, where "he training process" should indeed be corrected to "the training process", is well noted.
>
> We greatly appreciate your attention to this detail and will immediately correct this error in our next revision.
>
> ## For *Questions For Authors*:
>
> > I wonder compared to RAG, what is the pros and cons using the editing method.
>
> As mentioned earlier, RAG represents a parameter-preserving knowledge editing method that assists LLM generation through external knowledge retrieval rather than directly modifying the model's internal knowledge representations.
>
> Parameter-preserving methods (*e.g.*, RAG, SERAC, GRACE)  and parameter-modifying methods (*e.g.*, MEND, MEMIT, our proposed RLEdit) each have their advantages and limitations for different application scenarios.
>
> For **RAG**:
>
> * **Advantages**: RAG primarily targets knowledge-enhanced generation tasks, effectively supplementing information beyond LLM's knowledge base to produce more comprehensive and accurate outputs. It offers easy deployment, high computational efficiency, and enables dynamic knowledge updates.
>
> - **Limitations**: When target editing knowledge conflicts with existing knowledge stored in model parameters, RAG struggles to fundamentally alter LLM's internal cognition, potentially leading to inconsistent outputs or performance degradation. Additionally, RAG depends on external knowledge base quality and retrieval precision, increasing system complexity.
>
> For **Editing Methods**:
>
> - **Advantages**: These methods (like our RLEdit) fundamentally reshape model knowledge representations through direct parameter modification. Such modifications offer persistence, scalability, and end-to-end characteristics. The modified model can reflect updated knowledge without requiring additional modules, making them particularly suitable for knowledge correction and update tasks.
> - **Limitations**: Parameter-modifying methods typically demand higher computational resources, especially for large-scale models. Furthermore, multiple consecutive edits may lead to knowledge  interference and error accumulation, which is precisely the core problem our RLEdit attempts to mitigates.
>
> ******
>
> Thank you again for your feedback! If you have any further questions or suggestions regarding our methodology, experimental design, or potential application scenarios, we very much welcome continued in-depth discussion.
>
> Best regards
>
> Authors

---

### Decision · Program_Chairs · 2025-05-01

**Decision:**

Accept (poster)

**Comment:**

This paper introduces RLEdit a method that frames lifelong LLM editing as a reinforcement learning problem. RLEdit demonstrates significant advantages over existing approaches with a 59.24% improvement in effectiveness while requiring only 2.11% of computation time.

Key strengths include: its theoretically sound framework, successful extension of meta-learning to long sequence editing (up to 20k edits), strong performance across multiple models and datasets, and stability in maintaining LLM capabilities after extensive editing.

During the rebuttal phase, the authors addressed all reviewer concerns with additional experiments showing stable performance even after 20k edits, comparative analysis with parameter-preserving methods, and clear explanations of theoretical foundations, further strengthening the case for acceptance.